# Efficient Hierarchical Bayesian Inference for Spatio-temporal Regression Models in Neuroimaging

**Ali Hashemi**[1,2], **Yijing Gao**[3], **Chang Cai**[3,4], **Sanjay Ghosh**[3],
**Klaus-Robert Müller**[2,5,6,7], **Srikantan S. Nagarajan**[3], and **Stefan Haufe**[1,8,9,10]

[1]Uncertainty, Inverse Modeling and Machine Learning Group, Technische Universität Berlin, Germany.
[2]Machine Learning Group, Technische Universität Berlin, Germany.
[3]Department of Radiology and Biomedical Imaging, University of California, San Francisco, USA.
[4]National Engineering Research Center for E-Learning, Central China Normal University, China.
[5]BIFOLD – Berlin Institute for the Foundations of Learning and Data, Berlin, Germany.
[6]Department of Artificial Intelligence, Korea University, South Korea.
[7]Max Planck Institute for Informatics, Saarbrücken, Germany.
[8]Physikalisch-Technische Bundesanstalt, Berlin, Germany.
[9]Charité – Universitätsmedizin Berlin, Germany.
[10]Bernstein Center for Computational Neuroscience, Berlin, Germany.

## Abstract

Several problems in neuroimaging and beyond require inference on the parameters of multi-task sparse hierarchical regression models. Examples include M/EEG inverse problems, neural encoding models for task-based fMRI analyses, and climate science. In these domains, both the model parameters to be inferred and the measurement noise may exhibit a complex spatio-temporal structure. Existing work either neglects the *temporal* structure or leads to computationally demanding inference schemes. Overcoming these limitations, we devise a novel flexible hierarchical Bayesian framework within which the spatio-temporal dynamics of model parameters and noise are modeled to have Kronecker product covariance structure. Inference in our framework is based on majorization-minimization optimization and has guaranteed convergence properties. Our highly efficient algorithms exploit the intrinsic Riemannian geometry of temporal autocovariance matrices. For stationary dynamics described by Toeplitz matrices, the theory of circulant embeddings is employed. We prove convex bounding properties and derive update rules of the resulting algorithms. On both synthetic and real neural data from M/EEG, we demonstrate that our methods lead to improved performance.

## 1 Introduction

Probabilistic graphical models for regression problems, where both the model parameters and the observation noise can have complex spatio-temporal correlation structure, are prevalent in neuroscience, neuroimaging and beyond [1]. Example domains where these models are being applied include Gaussian process inference [2], sparse signal recovery and multi-task learning [3–6], array signal processing [7], fMRI analysis [8], climate science [9], computer hardware monitoring [10], and brain-computer interfaces (e.g. [11, 12]). The focus in this paper will be on brain source imaging (BSI) [13], i.e. the reconstruction of brain source activity from non-invasive magneto- or electroetoencephalography (M/EEG) sensors [14]. Although a number of generative models that encompass spatio-temporal correlations have been proposed within the BSI regression problem, inference solutions have either imposed specific simplifications and model constraints [15–17] or have ignored temporal correlations overall [5], and therefore not fully addressed the inherent spatio-temporal problem structure.

We propose a novel and efficient algorithmic framework using Type-II Bayesian regression that explicitly considers the spatio-temporal covariance structure in both model coefficients and observation noise. To this end, we focus on the M/EEG inverse problem of brain source reconstruction, for which we adopt a multi-task regression approach and formulate the source reconstruction problem as a probabilistic generative model with Gaussian scale mixture priors for sources with spatio-temporal prior covariance matrices that are expressed as a Kronecker product of separable spatial and temporal covariances. Their solutions are constrained to the Riemannian manifold of positive definite (P.D.) temporal covariance matrices. Exploiting the concept of geodesic convexity on the Riemannian manifold of covariance matrices with Kronecker structure, we are then able to derive robust, fast and efficient *majorization-minimization* (MM) optimization algorithms for model inference with provable convergence guarantees. In addition to deriving update rules for full-structural P.D. temporal covariance matrices, we also show that assuming a Toeplitz structure for the temporal covariance matrix results in computationally efficient update rules within the proposed MM framework.

## 2  Spatio-temporal generative model

Let us consider a multi-task linear regression problem, mathematically formulated as

$$\mathbf{Y}_g = \mathbf{L}\mathbf{X}_g + \mathbf{E}_g \quad \text{for } g = 1, \ldots, G , \tag{1}$$

in which a forward matrix, $\mathbf{L} \in \mathbb{R}^{M \times N}$, maps a set of coefficients or source components $\mathbf{X}_g \in \mathbb{R}^{N \times T}$ to measurements $\mathbf{Y}_g \in \mathbb{R}^{M \times T}$, with independent Gaussian noise $\mathbf{E}_g \in \mathbb{R}^{M \times T}$. $G$ denotes the number of sample blocks in the multi-task problem, while $N$, $M$, and $T$ denote the number of sources or coefficients, the number of sensors or observations, and the number of time instants, respectively. The problem of estimating $\{\mathbf{X}_g\}_{g=1}^G$ given $\mathbf{L}$ and $\{\mathbf{Y}_g\}_{g=1}^G$ can represent an inverse problem in physics, a multi-task regression problem in machine learning, or a multiple measurement vector (MMV) recovery problem in signal processing [18].

In the context of BSI, $\{\mathbf{Y}_g\}_{g=1}^G$ refers to M/EEG sensor measurement data, and $\{\mathbf{X}_g\}_{g=1}^G$ refers to brain source activity contributing to the observed sensor data. Here, $G$ can be defined as the number of epochs, trials or experimental tasks. The goal of BSI is to infer the underlying brain activity from the M/EEG measurements given the lead field matrix $\mathbf{L}$. In practice, $\mathbf{L}$ can be computed using discretization methods such as the finite element method for a given head geometry and known electrical conductivities using the quasi-static approximation of Maxwell's equations [19]. As the number of locations of potential brain sources is dramatically larger than the number of sensors, the M/EEG inverse problem is highly ill-posed, which can be dealt with by incorporating prior assumptions. Adopting a Bayesian treatment is useful in this capacity because it allows these assumptions to be made explicitly by specifying prior distributions for the model parameters. Inference can be performed either through Maximum-a-Posteriori (MAP) estimation (*Type-I Bayesian learning*) [20–22] or, when the model has unknown hyperparameters, through Type-II Maximum-Likelihood (ML) estimation (*Type-II Bayesian learning*) [23–25].

Here we focus on full spatio-temporal Type-II Bayesian learning, which assumes a family of prior distributions $p(\mathbf{X}_g | \mathbf{\Theta})$ parameterized by a set of hyperparameters $\mathbf{\Theta}$. We further assume that the spatio-temporal correlation structure of the brain sources can be modeled by a Gaussian $p(\mathbf{x}_g | \mathbf{\Gamma}, \mathbf{B}) \sim \mathcal{N}(\mathbf{0}, \mathbf{\Sigma}_0)$, where $\mathbf{x}_g = \text{vec}(\mathbf{X}_g^\top) \in \mathbb{R}^{NT \times 1}$ and where the covariance matrix $\mathbf{\Sigma}_0 = \mathbf{\Gamma} \otimes \mathbf{B}$ has Kronecker structure, meaning that the temporal and spatial correlations of the sources are modeled independently through matrices $\mathbf{\Gamma}$ and $\mathbf{B}$, where $\otimes$ stands for the Kronecker product. Note that the decoupling of spatial and temporal covariance is a neurophysiologically plausible assumption as neuronal sources originating from different parts of the brain can be assumed to share a similar autocorrelation spectrum. In this paper, we set $\mathbf{\Gamma} = \text{diag}(\boldsymbol{\gamma})$, $\boldsymbol{\gamma} = [\gamma_1, \ldots, \gamma_N]^\top$, and $\mathbf{B} \in \mathbb{S}_{++}^T$, where $\mathbb{S}_{++}^T$ denotes the set of positive definite matrices with size $T \times T$. Note that using a diagonal matrix $\mathbf{\Gamma}$ amounts to assuming that sources in different locations are independent. This assumption may be relaxed in our future work. On the other hand, modeling a full temporal covariance matrix $\mathbf{B}$ amounts to assuming non-stationary dynamics of the sources, which is appropriate in event-related experimental designs. To deal with the more stationary sources (ongoing brain activity), we will also constrain $\mathbf{B}$ to have Toeplitz structure later on. Based on these specifications, the prior distribution of the $i$-th brain source is modeled as $p((\mathbf{X}_g)_{i.} | \gamma_i, \mathbf{B}) \sim \mathcal{N}(0, \gamma_i \mathbf{B})$, for $i = 1, \ldots, N$, where $(\mathbf{X}_g)_{i.}$ denotes the $i$-th row of source matrix $\mathbf{X}_g$.

Analogous to the definition of the sources, we here also model the noise to have spatio-temporal structure. The matrix $\mathbf{E}_g \in \mathbb{R}^{M \times T}$ represents $T$ time instances of zero-mean Gaussian noise with full covariance, $\mathbf{\Sigma_e} = \mathbf{\Lambda} \otimes \mathbf{\Upsilon}$, where $\mathbf{e}_g = \text{vec}(\mathbf{E}_g^\top) \in \mathbb{R}^{MT \times 1} \sim \mathcal{N}(0, \mathbf{\Sigma_e})$, and where $\mathbf{\Lambda}$ and $\mathbf{\Upsilon}$ denote spatial and temporal noise covariance matrices, respectively. Here, we assume that the noise and sources share the same temporal structure, i.e., $\mathbf{\Upsilon} = \mathbf{B}$. We later investigate violations of this assumption in the simulation section. Analogous to the sources, we consider spatially independent noise characterized by diagonal spatial covariance $\mathbf{\Lambda} = \text{diag}(\boldsymbol{\lambda})$, $\boldsymbol{\lambda} = [\lambda_1, \ldots, \lambda_M]^\top$. We consider both the general, *heteroscedastic*, case in which the noise level may be different for each sensor as well as the *homoscedastic* case in which the noise level is uniform.

For later use, we also define augmented versions of the source and noise covariance matrices as well as the lead field. Specifically, we set $\mathbf{H}$ and $\mathbf{\Phi}$ so that $\mathbf{H} := [\mathbf{\Gamma}, \mathbf{0}; \mathbf{0}, \mathbf{\Lambda}]$, and $\mathbf{\Phi} := [\mathbf{L}, \mathbf{I}]$. These definitions allow us to unify source and noise covariance parameters within the single variable $\mathbf{H}$, which facilitates the concurrent estimation of both quantities. $\mathbf{\Phi}$ now plays the same role as the lead field $\mathbf{L}$, in the sense that it maps $\boldsymbol{\eta}_g := [\mathbf{x}_g^\top, \mathbf{e}_g^\top]$ to the measurements; i.e., $\mathbf{y}_g = \mathbf{\Phi}\boldsymbol{\eta}_g$. Figure 1-(a) illustrates a probabilistic graphical model summarizing the spatio-temporal generative model of our multi-task linear regression problem.

The MMV model Eq. (1) can be formulated equivalently in *single measurement vector (SMV)* form by vectorizing the spatio-temporal data matrices and using the Kronecker product as follows: $\mathbf{y}_g = \mathbf{D}\mathbf{x}_g + \mathbf{e}_g$, where $\mathbf{y}_g = \text{vec}(\mathbf{Y}_g^\top) \in \mathbb{R}^{MT \times 1}$, and $\mathbf{D} = \mathbf{L} \otimes \mathbf{I}_T$. In a Type-II Bayesian learning framework, the hyper-parameters of the spatio-temporal source model are optimized jointly with the model parameters $\{\boldsymbol{X}_g\}_{g=1}^G$. In our case, these hyper-parameters comprise the unknown source, noise and temporal covariance matrices, i.e., $\mathbf{\Theta} = \{\mathbf{\Gamma}, \mathbf{\Lambda}, \mathbf{B}\}$.

## 3 Proposed method — full Dugh

The unknown parameters $\mathbf{\Gamma}$, $\mathbf{\Lambda}$, and $\mathbf{B}$ can be optimized in an alternating iterative process. Given initial estimates, the posterior distribution of the sources is a Gaussian of the form $p(\mathbf{x}_g|\mathbf{y}_g, \mathbf{\Gamma}, \mathbf{\Lambda}, \mathbf{B}) \sim \mathcal{N}(\bar{\mathbf{x}}_g, \mathbf{\Sigma_x})$, whose mean and covariance are defined as follows:

$$\bar{\mathbf{x}}_g = \mathbf{\Sigma}_0 \mathbf{D}^\top (\mathbf{\Lambda} \otimes \mathbf{B} + \mathbf{D}\mathbf{\Sigma}_0\mathbf{D}^\top)^{-1}\mathbf{y}_g = \mathbf{\Sigma}_0\mathbf{D}^\top \tilde{\mathbf{\Sigma}}_{\mathbf{y}}^{-1}\mathbf{y}_g \ , \tag{2}$$

$$\mathbf{\Sigma_x} = \mathbf{\Sigma}_0 - \mathbf{\Sigma}_0\mathbf{D}^\top \tilde{\mathbf{\Sigma}}_{\mathbf{y}}^{-1}\mathbf{D}\mathbf{\Sigma}_0 \ , \tag{3}$$

where $\mathbf{\Sigma_y} = \mathbf{L}\mathbf{\Gamma}\mathbf{L}^\top + \mathbf{\Lambda}$, and where $\tilde{\mathbf{\Sigma}}_{\mathbf{y}} = \mathbf{\Sigma_y} \otimes \mathbf{B}$ denotes the spatio-temporal statistical model covariance matrix. The estimated posterior parameters $\bar{\mathbf{x}}_g$ and $\mathbf{\Sigma_x}$ are then in turn used to update $\mathbf{\Gamma}$, $\mathbf{\Lambda}$, and $\mathbf{B}$ as the minimizers of the negative log of the marginal likelihood $p(\mathbf{Y}|\mathbf{\Gamma}, \mathbf{\Lambda}, \mathbf{B})$, which is given by:

$$\mathcal{L}_{\text{kron}}(\mathbf{\Gamma}, \mathbf{\Lambda}, \mathbf{B}) = T \log|\mathbf{\Sigma_y}| + M \log|\mathbf{B}| + \frac{1}{G}\sum_{g=1}^G \text{tr}(\mathbf{\Sigma_y}^{-1}\mathbf{Y}_g\mathbf{B}^{-1}\mathbf{Y}_g^\top) \ , \tag{4}$$

where $|\cdot|$ denotes the determinant of a matrix. A detailed derivation is provided in Appendix A. Given the solution of the hyperparameters $\mathbf{\Gamma}$, $\mathbf{\Lambda}$, and $\mathbf{B}$, the posterior source distribution is obtained by plugging these estimates into Eqs. (2)–(3). This process is repeated until convergence.

The challenge in this high-dimensional inference problem is to find (locally) optimal solutions to Eq. (4), which is a non-convex cost function, in adequate time. Here we propose a novel efficient algorithm, which is able to do so. Our algorithm thus learns the full spatio-temporal correlation structure of sources and noise. Hashemi et al. [26] have previously demonstrated that *majorization-minimization* (MM) [27] is a powerful non-linear optimization framework that can be leveraged to solve similar Bayesian Type-II inference problems. Here we extend this work to our spatio-temporal setting. Building on the idea of majorization-minimization, we construct convex surrogate functions that *majorize* $\mathcal{L}_{\text{kron}}(\mathbf{\Gamma}, \mathbf{\Lambda}, \mathbf{B})$ in each iteration of the proposed algorithm. Then, we show the minimization equivalence between the constructed majorizing functions and Eq. (4). These results are presented in theorems 1 and 3. Theorems 2 and 4 propose an efficient alternating optimization algorithm for solving $\mathcal{L}_{\text{kron}}(\mathbf{\Gamma}, \mathbf{\Lambda}, \mathbf{B})$, which leads to update rules for the spatial and temporal covariance matrices $\mathbf{\Gamma}$ and $\mathbf{B}$ as well as the noise covariance matrix $\mathbf{\Lambda}$.

Starting with the estimation of the temporal covariance based on the current source estimate, we can state the following theorem:

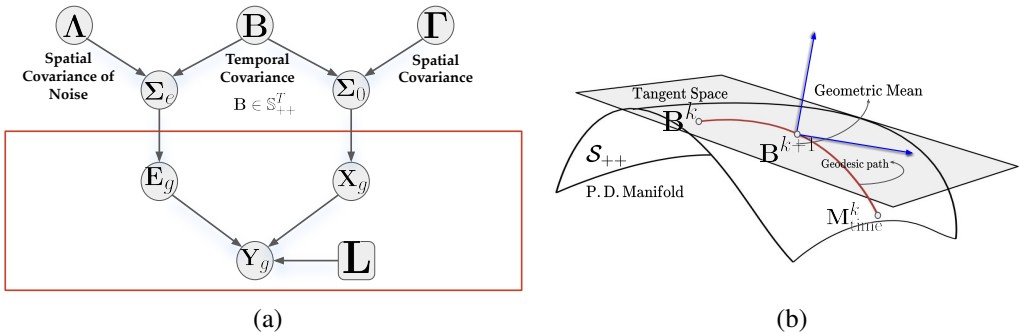

(a)             (b)

Figure 1: (a) Probabilistic graphical model for the multi-task linear regression problem. (b) Geometric representation of the geodesic path between the pair of matrices $\{\mathbf{B}^k, \mathbf{M}_{\text{time}}^k\}$ on the P.D. manifold and the geometric mean between them, which is used to update $\mathbf{B}^{k+1}$.

**Theorem 1.** *Optimizing the non-convex Type-II cost function in Eq. (4), $\mathcal{L}_{kron}(\boldsymbol{\Gamma}, \boldsymbol{\Lambda}, \mathbf{B})$, with respect to the temporal covariance matrix $\mathbf{B}$ is equivalent to optimizing the following convex surrogate function, which* majorizes *Eq. (4):*

$$\mathcal{L}_{\text{conv}}^{\text{time}}(\boldsymbol{\Gamma}^k, \boldsymbol{\Lambda}^k, \mathbf{B}) = \text{tr}\left((\mathbf{B}^k)^{-1}\mathbf{B}\right) + \text{tr}(\mathbf{M}_{\text{time}}^k \mathbf{B}^{-1}), \tag{5}$$

*where $\mathbf{M}_{\text{time}}^k$ is defined as*

$$\mathbf{M}_{\text{time}}^k := \frac{1}{MG}\sum_{g=1}^{G} \mathbf{Y}_g^\top \left(\boldsymbol{\Sigma}_{\mathbf{y}}^k\right)^{-1} \mathbf{Y}_g \,, \tag{6}$$

*and where $\boldsymbol{\Gamma}^k$, $\boldsymbol{\Lambda}^k$, and $\boldsymbol{\Sigma}_{\mathbf{y}}^k$ denote the source, noise and statistical model covariance matrices at the $k$-th iteration, which are treated as constants when optimizing over $\mathbf{B}$.*

*Proof.* A detailed proof is provided in Appendix B. □

The solution of Eq. (5) can be obtained by taking advantage of the Riemannian geometry of the search space. The following theorem holds.

**Theorem 2.** *The cost function $\mathcal{L}_{\text{conv}}^{\text{time}}(\boldsymbol{\Gamma}^k, \boldsymbol{\Lambda}^k, \mathbf{B})$ is strictly geodesically convex with respect to the P.D. manifold and its minimum with respect to $\mathbf{B}$ can be attained by iterating the following update rule until convergence:*

$$\mathbf{B}^{k+1} \leftarrow (\mathbf{B}^k)^{1/2}\left((\mathbf{B}^k)^{-1/2}\mathbf{M}_{\text{time}}^k(\mathbf{B}^k)^{-1/2}\right)^{1/2}(\mathbf{B}^k)^{1/2} \,. \tag{7}$$

*Proof.* A detailed proof is provided in Appendix C. □

**Remark 1.** *A geometric interpretation of the update rule (7) is that it finds the geometric mean between a spatially whitened version of the empirical temporal covariance matrix, $\mathbf{M}_{\text{time}}^k$, and the model temporal covariance matrix from the previous iteration, $\mathbf{B}^k$. A geometric representation of the geodesic path between the pair of matrices $\{\mathbf{B}^k, \mathbf{M}_{\text{time}}^k\}$ on the P.D. manifold and the geometric mean between them, representing the update for $\mathbf{B}^{k+1}$, is provided in Figure 1-(b).*

We now derive an update rule for the spatial source and noise covariance matrices using an analogous approach. To this end, we first construct a convex surrogate function that *majorizes* $\mathcal{L}_{\text{kron}}(\boldsymbol{\Gamma}, \boldsymbol{\Lambda}, \mathbf{B})$ in each iteration of the optimization algorithm by considering augmented variables comprising both the source and noise covariances, i.e., $\mathbf{H} := [\boldsymbol{\Gamma}, \mathbf{0}; \mathbf{0}, \boldsymbol{\Lambda}]$, and $\boldsymbol{\Phi} := [\mathbf{L}, \mathbf{I}]$. The following theorem holds.

**Theorem 3.** *Minimizing the non-convex Type-II cost function in Eq. (4), $\mathcal{L}_{kron}(\boldsymbol{\Gamma}, \boldsymbol{\Lambda}, \mathbf{B})$, with respect to the spatial covariance matrix $\mathbf{H}$ is equivalent to minimizing the following convex surrogate function, which* majorizes *Eq. (4):*

$$\mathcal{L}_{\text{conv}}^{\text{space}}(\boldsymbol{\Gamma}, \boldsymbol{\Lambda}, \mathbf{B}^k) = \mathcal{L}_{\text{conv}}^{\text{space}}(\mathbf{H}, \mathbf{B}^k) = \text{tr}\left(\boldsymbol{\Phi}^\top(\boldsymbol{\Sigma}_{\mathbf{y}}^k)^{-1}\boldsymbol{\Phi}\mathbf{H}\right) + \text{tr}\left(\mathbf{M}_{\text{SN}}^k \mathbf{H}^{-1}\right) \,, \tag{8}$$

where $\mathbf{M}_{\mathrm{SN}}^k$ is defined as

$$\mathbf{M}_{\mathrm{SN}}^k := \mathbf{H}^k \boldsymbol{\Phi}^\top (\boldsymbol{\Sigma}_{\mathbf{y}}^k)^{-1} \mathbf{M}_{\mathrm{space}}^k (\boldsymbol{\Sigma}_{\mathbf{y}}^k)^{-1} \boldsymbol{\Phi} \mathbf{H}^k \text{, with}$$

$$\mathbf{M}_{\mathrm{space}}^k := \frac{1}{TG} \sum_{g=1}^G \mathbf{Y}_g (\mathbf{B}^k)^{-1} \mathbf{Y}_g^\top \ , \tag{9}$$

and where $\mathbf{B}^k$ denotes the temporal model covariance matrix estimated in the $k$-th iteration.

*Proof.* A detailed proof is provided in Appendix D. $\qquad\square$

While, in principle, update rules for full-structured spatial source and noise covariances may be conceived, in analogy to the ones presented for the temporal covariance, we here restrict ourselves to the discussion of diagonal spatial covariances. Note, though, that such a choice of prior does not prohibit the reconstruction of parameters with more complex correlation structure. We have $\mathbf{H} = \mathrm{diag}(\mathbf{h})$, $\mathbf{h} = [\gamma_1, \ldots, \gamma_N, \sigma_1^2, \ldots, \sigma_M^2]^\top$, and $\boldsymbol{\Sigma}_{\mathbf{y}} = \boldsymbol{\Phi} \mathbf{H} \boldsymbol{\Phi}^\top$. The update rule for $\mathbf{H}$ then takes a simple form, as stated in the following theorem:

**Theorem 4.** *The cost function $\mathcal{L}_{\mathrm{conv}}^{\mathrm{space}}(\mathbf{H}, \mathbf{B}^k)$ is convex in $\mathbf{h}$, and its minimum with respect to $\mathbf{h}$ can be obtained according to the following closed-form update rule, which concurrently estimates the scalar source variances and heteroscedastic noise variances:*

$$\mathbf{H}^{k+1} = \mathrm{diag}(\mathbf{h}^{k+1}), \ h_i^{k+1} \leftarrow \sqrt{\frac{g_i^k}{z_i^k}} \quad \text{for } i = 1, \ldots, N+M \text{ , where} \tag{10}$$

$$\mathbf{g} := \mathrm{diag}(\mathbf{M}_{\mathrm{SN}}^k) \tag{11}$$

$$\mathbf{z} := \mathrm{diag}(\boldsymbol{\Phi}^\top (\boldsymbol{\Sigma}_{\mathbf{y}}^k)^{-1} \boldsymbol{\Phi}) \tag{12}$$

*Proof.* A detailed proof can be found in Appendix E. $\qquad\square$

The final estimate of the posterior distribution (2) can be obtained by starting from any initial guess for the model parameters, $\mathbf{H}^0 = \{\boldsymbol{\Gamma}^0, \boldsymbol{\Lambda}^0\}$ and $\mathbf{B}^0$, and iterating between update rules (7), (10), and (2) until convergence. We call the resulting algorithm *full Dugh*. Convergence is shown in the following theorem:

**Theorem 5.** *Optimizing the non-convex ML cost function Eq. (4) with alternating update rules for $\mathbf{B}$ and $\mathbf{H} = \{\boldsymbol{\Gamma}, \boldsymbol{\Lambda}\}$ in Eq. (7) and Eq. (10) defines an MM algorithm, which is guaranteed to converge to a stationary point.*

*Proof.* A detailed proof can be found in Appendix G. $\qquad\square$

## 4 Efficient inference for stationary temporal dynamics — thin Dugh

With full Dugh, we could present a general inference method for data with full temporal covariance structure. This algorithm may be used for non-stationary data, for which the covariance between two time samples depends on the absolute position of the two samples within the studied time window. An example is the reconstruction of brain responses to repeated external stimuli from M/EEG (event-related electrical potential or magnetic field) data, where data blocks are aligned to the stimulus onset. In other realistic settings, however, stationarity can be assumed in the sense that the covariance between two samples only depends on their relative position (distance) to each other but not to any external trigger. An example is the reconstruction of ongoing brain dynamics in absence of known external triggers. This situation can be adequately modeled using temporal covariance matrices with Toeplitz structure. In the following, we devise an efficient extension of full Dugh for that setting.

Temporal correlation has been incorporated in various brain source imaging models through different priors [see 13, 28, and references therein]. According to [29], the temporal dynamics of neuronal populations can be approximated by a first order auto-regressive (AR(1)) model of the form $x_n(t + 1) = \beta x_n(t) + \sqrt{1 - \beta^2} \xi_n(t)$, $n = 1, \ldots, N; t = 1, \ldots, T$ with AR coefficient $\beta \in (-1, 1)$ and innovation noise $\xi_n(t)$. It can be shown that the temporal correlation matrix corresponding to this AR

model has Toeplitz structure, $\mathbf{B}_{i,j} = \beta^{|i-j|}$ [30]. Consequently, we now constrain the cost function Eq. (5) to the set of Toeplitz matrices:

$$\mathbf{B}^{k+1} = \underset{\mathbf{B} \in \mathcal{B}, \ \mathbf{H} = \mathbf{H}^k}{\arg\min} \ \mathrm{tr}((\mathbf{B}^k)^{-1}\mathbf{B}) + \mathrm{tr}(\mathbf{M}_{\mathrm{time}}^k \mathbf{B}^{-1}) \ , \tag{13}$$

where set $\mathcal{B}$ denotes the set of real-valued positive-definite Toeplitz matrices of size $T \times T$.

In order to be able to derive efficient update rules for this setting, we bring $\mathbf{B}$ into a diagonal form. The following proposition holds.

**Proposition 1.** *Let $\mathbf{P} \in \mathbb{R}^{L \times L}$ with $L > T$, be the circulant embedding of matrix $\mathbf{B} \in \mathcal{B}^L$, where $\mathcal{B}^L$ is assumed to be the subset of $\mathcal{B}$ that guarantees that $\mathbf{P}$ is a real-valued circulant P.D. matrix. Then the Toeplitz matrix $\mathbf{B}$ can be diagonalized as follows:*

$$\mathbf{B} = \mathbf{Q}\mathbf{P}\mathbf{Q}^H \quad with \quad \mathbf{Q} = [\mathbf{I}_M, \mathbf{0}]\mathbf{F}_L, \ and \tag{14}$$

$$[\mathbf{F}_L]_{m,l} = \frac{1}{\sqrt{L}} e^{\frac{i2\pi(l-1)}{L}(m-1)} \ , \tag{15}$$

*where $\mathbf{P} = \mathrm{diag}(\mathbf{p}) = \mathrm{diag}(p_0, p_1, \ldots, p_{L-1})$ is a diagonal matrix. The main diagonal coefficients of $\mathbf{P}$ are given as the normalized discrete Fourier transform (DFT), represented by the linear operator $\mathbf{F}_L$, of the first row of $\mathbf{B}$: $\mathbf{p} = \mathbf{F}_L \mathbf{B}_{1.}$ (Note that a Toeplitz matrix can be represented by its first row or column, $\mathbf{B}_{1.}$).*

*Proof.* This is a direct implication of the fact that Toeplitz matrices can be embedded into circulant matrices of larger size [31] and the result that circulant matrices can be approximately diagonalized using the Fourier transform [32]. Further details can be found in Appendix H. □

The solution of Eq. (13) can be obtained by direct application of Proposition 1. The results is summarized in the following theorem:

**Theorem 6.** *The cost function Eq. (13) is convex in $\mathbf{p}$, and its minimum with respect to $\mathbf{p}$ can be obtained by iterating the following closed-form update rule until convergence:*

$$p_l^{k+1} \leftarrow \sqrt{\frac{\hat{g}_l^k}{\hat{z}_l^k}} \ for \ l = 1, \ldots, L \ , where \tag{16}$$

$$\hat{\mathbf{g}} := \mathrm{diag}(\mathbf{P}^k \mathbf{Q}^H (\mathbf{B}^k)^{-1} \mathbf{M}_{\mathrm{time}}^k (\mathbf{B}^k)^{-1} \mathbf{Q}\mathbf{P}^k) \tag{17}$$

$$\hat{\mathbf{z}} := \mathrm{diag}(\mathbf{Q}^H (\mathbf{B}^k)^{-1} \mathbf{Q}) \ . \tag{18}$$

*Proof.* A detailed proof is provided in Appendix I. □

### 4.1 Efficient computation of the posterior

Given estimates of the spatial and temporal covariance matrices, we can efficiently compute the posterior mean by exploiting their intrinsic diagonal structure.

**Theorem 7.** *Given the diagonalization $\mathbf{B} = \mathbf{Q}\mathbf{P}\mathbf{Q}^H$ of the temporal correlation matrix and the eigenvalue decomposition $\mathbf{L}\mathbf{\Gamma}\mathbf{L}^\top = \mathbf{U_x}\mathbf{D_x}\mathbf{U_x}^\top$ of $\mathbf{L}\mathbf{\Gamma}\mathbf{L}^\top$, where $\mathbf{D_x} = \mathrm{diag}(d_1, \ldots, d_M)$, the posterior mean is efficiently computed as*

$$\bar{\mathbf{x}}_g = (\mathbf{\Gamma} \otimes \mathbf{B}) \mathbf{D}^\top \tilde{\mathbf{\Sigma}}_\mathbf{y}^{-1} \mathbf{y}_g = \mathrm{tr}\left(\mathbf{Q}\mathbf{P}\left(\mathbf{\Pi} \odot \mathbf{Q}^H \mathbf{Y}_g^\top \mathbf{U_x}\right)\left(\mathbf{U_x}^\top \mathbf{L}\mathbf{\Gamma}^\top\right)\right) \ , \tag{19}$$

*where $\odot$ denotes the Hadamard product between the corresponding elements of two matrices of equal size. In addition, $\mathbf{\Pi}$ is defined as follows: $[\mathbf{\Pi}]_{l,m} = \frac{1}{\sigma_m^2 + p_l d_m}$ for $l = 1, \ldots, L$ and $m = 1, \ldots, M$.*

*Proof.* A detailed proof is provided in Appendix J. □

The resulting algorithm, obtained by iterating between update rules (16), (10) and (19), is called *thin Dugh* (as opposed to *full Dugh* introduced above).

# 5 Simulations

We present two sets of experiments to assess the performance of our proposed methods. In the first experiment, we compare the reconstruction performance of the proposed Dugh algorithm variants to that of Champagne [33] and two other competitive methods – eLORETA [34] and S-FLEX [35] – for a range of SNR levels, numbers of time samples, and orders of AR coefficients. In the second experiment, we test the impact of model violations on the temporal covariance estimation. All experiments are performed using Matlab on a machine with a 2.50 GHz Intel(R) Xeon(R) Platinum 8160 CPU. The computational complexity of each method in terms of the average running time in units of seconds for 1000 iterations is as follows: *Full Dugh*: 67.094s, *Thin Dugh*: 62.289s, *Champagne*: 1.533s, *eLORETA*: 2.653s, and *S-FLEX*: 20.963s. The codes are publicly available at https://github.com/AliHashemi-ai/Dugh-NeurIPS-2021.

## 5.1 Pseudo-EEG signal generation and benchmark comparison

We simulate a sparse set of $N_0 = 3$ active sources placed at random locations on the cortex. To simulate the electrical neural activity, we sample time series of length $T \in \{10, 20, 50, 100\}$ from a univariate linear autoregressive AR(P) process. We use stable AR systems of order $P \in \{1, 2, 5, 7\}$. The resulting source distribution is then projected to the EEG sensors, denoted by $\mathbf{Y}^{\text{signal}}$, using a realistic lead field matrix, $\mathbf{L} \in \mathbb{R}^{58 \times 2004}$. We generate $\mathbf{L}$ using the New York Head model [36] taking into account the realistic anatomy and electrical tissue conductivities of an average human head. Finally, we add Gaussian white noise to the sensor space signal. Note that the simulated noise and source time courses do not share a similar temporal structure here – sources are modeled with a univariate autoregressive AR(P) process while a temporally white Gaussian distribution is used for modeling noise. Thus, we could assess the robustness of our proposed method under violation of the model assumption that the temporal structure of sources and noise is similar. The resulting noise matrix $\mathbf{E} = [\mathbf{e}(1), \ldots, \mathbf{e}(T)]$ is first normalized and then added to the signal matrix $\mathbf{Y}^{\text{signal}}$ as follows: $\mathbf{Y} = \mathbf{Y}^{\text{signal}} + \frac{(1-\alpha)\|\mathbf{Y}^{\text{signal}}\|_F}{\alpha\|\mathbf{E}\|_F}\mathbf{E}$, where $\alpha$ determines the signal-to-noise ratio (SNR) in sensor space. Precisely, SNR is defined as $\text{SNR} = 20\log_{10}\left(\alpha/1-\alpha\right)$. In this experiment the following values of $\alpha$ are used: $\alpha \in \{0.55, 0.65, 0.7, 0.8\}$, which correspond to the following SNRs: $\text{SNR} \in \{1.7, 5.4, 7.4, 12\}$ (dB). Interested readers can refer to Appendix K and [37] for further details on the simulation framework. We quantify the performance of all algorithms using the *earth mover's distance* (EMD) [21, 38] and the maximal correlation between the time courses of the simulated and the reconstructed sources (TCE). Each simulation is carried out 100 times using different instances of $\mathbf{X}$ and $\mathbf{E}$, and the mean and standard error of the mean (SEM) of each performance measure across repetitions is calculated.

In order to investigate the impact of model violations on the temporal covariance estimation, we generate a random Gaussian source matrix, $\mathbf{X} \in \mathbb{R}^{2004 \times 30 \times G}$ representing the brain activity of 2004 brain sources at 30 time instances for different numbers of trials $G \in 10, 20, 30, 40, 50$. In all trials, sources are randomly sampled from a zero-mean normal distribution with spatio-temporal covariance matrix $\mathbf{\Gamma} \otimes \mathbf{B}$, where $\mathbf{B} \in \mathbb{R}^{30 \times 30}$ is either a full-structural random PSD matrix or a Toeplitz matrix with $\beta = 0.8$. Gaussian noise $\mathbf{E}$ sharing the same temporal covariance with the sources is added to the measurements $\mathbf{Y} = \mathbf{L}\mathbf{X}$, so that the overall SNR is 0 dB. We evaluate the accuracy of the temporal covariance reconstruction using Thin and full Dugh. The performance is evaluated using two measures: Pearson correlation between the original and reconstructed temporal covariance matrices, $\mathbf{B}$ and $\hat{\mathbf{B}}$, denoted by $r(\mathbf{B}, \hat{\mathbf{B}})$, and the normalized mean squared error (NMSE) defined as: $\text{NMSE} = \|\hat{\mathbf{B}} - \mathbf{B}\|_F^2/\|\mathbf{B}\|_F^2$. The similarity error is defined as: $1 - r(\mathbf{B}, \hat{\mathbf{B}})$. Note that NMSE measures the reconstruction at the true scale of the temporal covariance; while $r(\mathbf{B}, \hat{\mathbf{B}})$ is scale-invariant and hence only quantifies the overall structural similarity between simulated and estimated noise covariance matrices.

In Figure 2, we show the source reconstruction performance (mean $\pm$ SEM) of the four different brain source imaging schemes, namely thin Dugh, Champagne, eLORETA and S-FLEX. We notice that Dugh achieves superior performance in terms of EMD metric, whereas it is competitive in terms of TCE. Note that since thin Dugh incorporates the temporal structure of the sources into the inference scheme, its performance with respect to EMD and TCE can be significantly improved by increasing the number of time samples. Figure 3 demonstrates the estimated temporal covariance matrix obtained from our two proposed spatio-temporal learning schemes, namely thin and full Dugh, indicated by cyan and magenta colors, respectively. The upper panel illustrates the reconstruction results for a

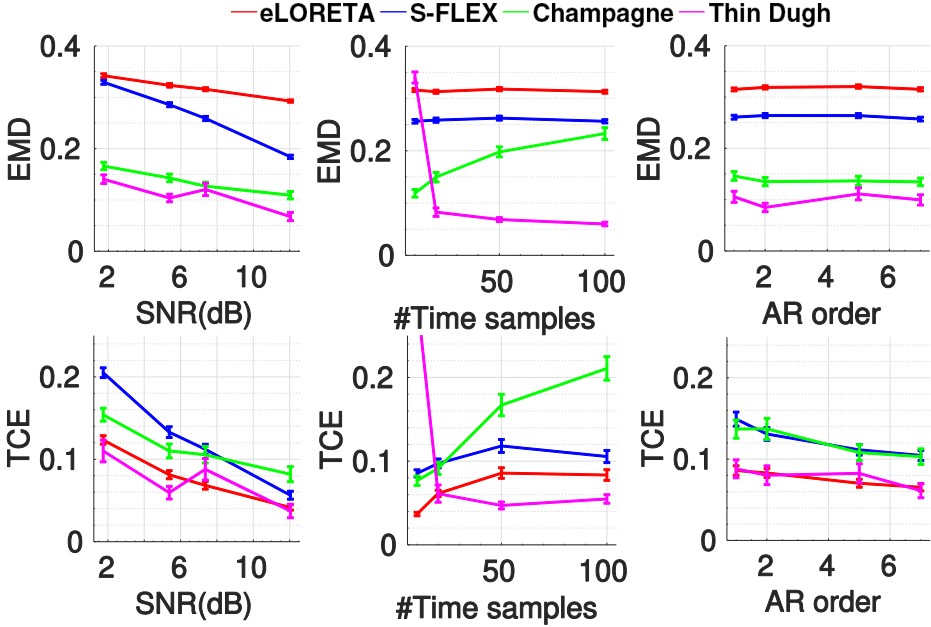

Figure 2: Source reconstruction performance (mean ± SEM) of the four different brain source imaging schemes, namely Thin Dugh, Champagne, eLORETA, and S-FLEX, for data generated by a realistic lead field matrix. Performance is assessed for different settings including a wide range of SNRs, different numbers of time samples, and different AR model orders. Performance is evaluated in terms of the earth mover's distance (EMD) and time-course correlation error (TCE) between each simulated source and the reconstructed source with highest maximum absolute correlation.

setting where the ground-truth temporal covariance matrix has full structure, while the lower panel shows a case with Toeplitz temporal covariance matrix structure. It can be seen that *full Dugh* can better capture the overall structure of ground truth full-structure temporal covariance as evidenced by lower NMSE and similarity errors compared to *thin Dugh* that is only able to recover a Toeplitz matrix. As can be expected, the behavior is reversed in the lower-panel of Figure 3, where the ground truth temporal covariance matrix is indeed Toeplitz. Full Dugh, however, still provides reasonable performance in terms of NMSE and similarity error, even though it estimates a full-structure temporal covariance matrix. Finally, the performance of both learning schemes are significantly improved by increasing the number of trials.

# 6 Analysis of real MEG and EEG data recordings

We now demonstrate the performance of the novel algorithms on two MEG datasets (see Figure 4 and 5) and one EEG dataset (see Appendix L). All participants provided informed written consent prior to study participation and received monetary compensation for their participation. The studies were approved by the University of California, San Francisco Committee on Human Research. The MEG datasets from two female subjects comprised five trials of visual or auditory stimulus presentation, where the goal was to reconstruct cortical activity reflecting auditory and visual processing. No prior work has shown success in reconstruction in such extreme low SNR data.

Figure 4 shows the reconstructed sources for auditory evoked fields (AEF) from a representative subject using eLORETA, MCE, thin and full Dugh. In this case, we tested the reconstruction performance of all algorithms with the number of trials limited to 5. As Figure 4 demonstrates, the reconstructions of thin and full Dugh both show focal sources at the expected locations of the auditory cortex. Limiting the number of trials to as few as 5 does not negatively influence the reconstruction result of Dugh methods, while it severely affects the reconstruction performance of competing methods. For the visual evoked field (VEF) data in Figure 5, thin Dugh was able to reconstruct two sources in the visual cortex with their corresponding time-courses demonstrating an early peak at 100 ms and a later peak around 200 ms (similar performance was observed for full Dugh).

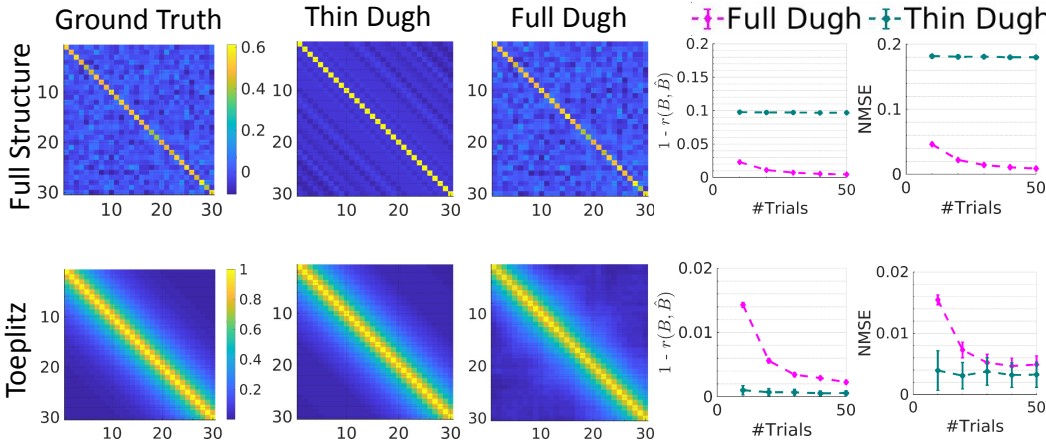

Figure 3: Accuracy of the temporal covariance matrix reconstruction incurred by two different temporal learning approaches, namely thin (cyan) and full (magenta) Dugh, assuming Toeplitz and full temporal covariance structure, respectively. The ground-truth temporal covariance matrix (first column) has full covariance structure in the first row, and Toeplitz structure in the second row. Performance is assessed in terms of the Pearson correlation between the entries of the original and reconstructed temporal covariance matrices, $\mathbf{B}$ and $\hat{\mathbf{B}}$, denoted by $r(\mathbf{B}, \hat{\mathbf{B}})$. Shown is the similarity error $1 - r(\mathbf{B}, \hat{\mathbf{B}})$ (forth column). Further, the normalized mean squared error (NMSE) between $\mathbf{B}$ and $\hat{\mathbf{B}}$, defined as $\text{NMSE} = ||\hat{\mathbf{B}} - \mathbf{B}||_F^2 / ||\mathbf{B}||_F^2$ is reported (fifth column).

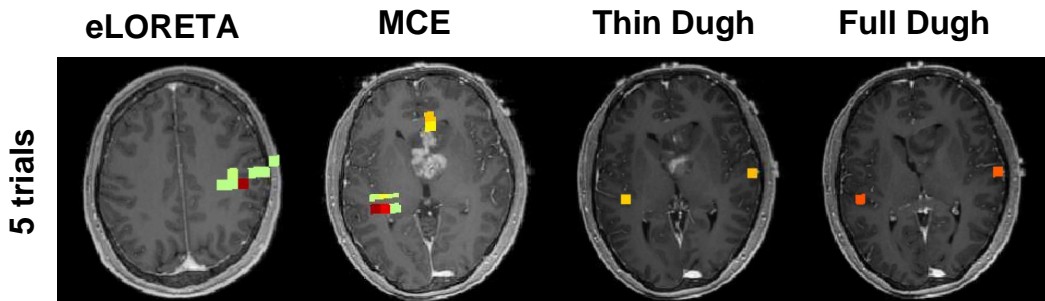

Figure 4: Analysis of auditory evoked fields (AEF) of one representative subject using eLORETA, MCE, thin and full Dugh. As it can be seen, thin and full Dugh can correctly localize bilateral auditory activity to Heschl's gyrus, which is the characteristic location of the primary auditory cortex, with as few as 5 trials. In this challenging setting, all competing methods show inferior performance.

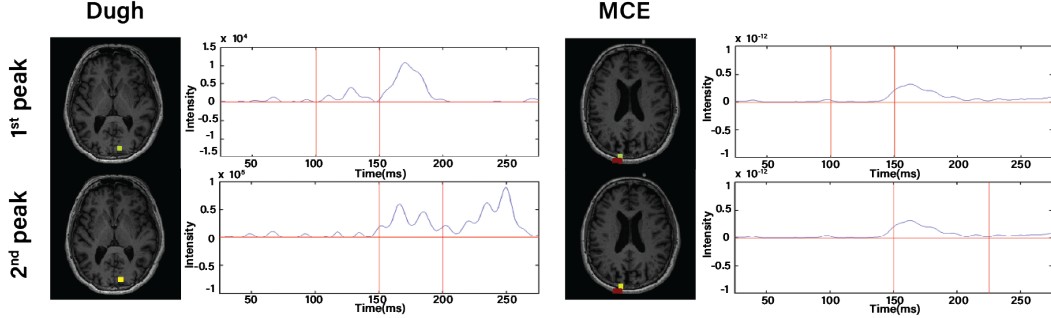

Figure 5: Spatial maps and corresponding time-courses reconstructed from five trials of visual stimulus presentation using Dugh and MCE. Distinct activity in visual cortex was reconstructed.

# 7 Discussion

Inverse modeling is a challenging problem that can be cast as a multi-task regression problem. Machine learning-based methods have contributed by systematically incorporating prior information into such models (e.g. [13, 25, 39–41]). While, so far, explicit models of spatio-temporal dynamics are rare, we contribute in this paper by deriving efficient inference algorithms for regression models with spatio-temporal dynamics in model parameters and noise. Specifically, we employ separable Gaussian distributions using Kronecker products of temporal and spatial covariance matrices. We assume sparsity and independent activations in the spatial domain, while the temporal covariance is modeled to have either full or Toeplitz structure. The proposed Dugh framework is encompassing efficient optimization algorithms for jointly estimating these distributions within a hierarchical Bayesian inference and MM optimization framework. Interestingly, we could theoretically prove convergence for the proposed update rules yielding estimates of the spatial and temporal covariances. In careful simulation studies, we have demonstrated that the inclusion of both spatial and temporal model parameters in the model indeed leads to significantly improved reconstruction performance. Finally, the utility of our algorithms is showcased in challenging real-world applications by reconstructing expected sources from real M/EEG data based on very few experimental trials.

The M/EEG BSI problem has been previously framed as a multi-task regression problem in the context of Type-I learning methods [5, 42]. Bertrand et al. [5] proposed a method to extend the group Lasso class of algorithms to the multi-task learning case. Their method shows promising results for large numbers of trials but is not competitive for smaller sample sizes, presumably due to not modeling the temporal structure of the sources. In contrast, Dugh is a Type-II method that learns the full spatio-temporal prior source distribution as part of the model fitting. Note that the assumption of spatially independent noise made here can be readily dropped, as in [43]. A number of alternative approaches have been proposed to estimate the spatio-temporal correlation structure of the *noise* [44–48]. These works, however, do not estimate the noise characteristics as part of the source reconstruction problem on the same data but require separate noise recordings. Our proposed algorithm substantially differs in this respect, as it learns the full noise covariance jointly with the brain source distribution. This joint estimation perspective is very much in line with the end-to-end learning philosophy as opposed to a step-wise independent estimation process that can give rise to error accumulation.

With respect to the limits of our Dugh framework, we would like to note that using the same temporal correlation prior for noise and sources is a potentially restricting assumption in our modeling. It is made here to achieve tractable inference due to the fact that the spatio-temporal statistical model covariance matrix can be formulated in a separable form, i.e., $\tilde{\Sigma}_{\mathbf{y}} = \Sigma_{\mathbf{y}} \otimes \mathbf{B}$. Although we have demonstrated empirically that the reconstruction results of our proposed learning schemes are fairly robust against violating this assumption, this constraint may be further relaxed by exploiting eigenvalue decomposition techniques presented in [2, 49]. Another potentially limiting assumption in our model is to assume Gaussian distributions for the sources and noise. Although Gaussian priors are commonly justified, the extension of our framework to heavy-tailed noise distributions, which are more robust to outliers, is a direction of future work. Regarding the societal impact of this work, we note that we intend to solve inverse problems that have non-unique solutions, which depend on explicit or implicit assumptions of priors. These have applications in basic, clinical, and translational neuroscience imaging studies. Our use of a hierarchical empirical Bayesian framework allows for explicit specifications of the priors in our model that are learned from data, and a more clear interpretation of the reconstructions with reduced bias. Nevertheless, in our algorithms, we assume sparse spatial priors, and in scenarios where this assumption embodied in the priors is incorrect, the resulting reconstructions will be inaccurate. Users and neuroscientists will need to be cognizant of these issues.

In conclusion, we could derive novel flexible hierarchical Bayesian algorithms for multi-task regression problems with spatio-temporal covariances. Incorporating prior knowledge, namely, constraining the solutions to Riemannian manifolds, and using ideas of geodesic convexity, circulant embeddings, and majorization-minimization, we derive inference update rules and prove convergence guarantees. The proposed Dugh algorithms show robust and competitive performance both on synthetic and real neural data from M/EEG recordings and thus contribute to a well-founded solution to complex inverse problems in neuroscience.

## Acknowledgments and Disclosure of Funding

This result is part of a project that has received funding from the European Research Council (ERC) under the European Union's Horizon 2020 research and innovation programme (Grant agreement No. 758985). AH acknowledges scholarship support from the Machine Learning/Intelligent Data Analysis research group at Technische Universität Berlin. He further wishes to thank the Charité – Universitätsmedizin Berlin, the Berlin Mathematical School (BMS), and the Berlin Mathematics Research Center MATH+ for partial support. CC was supported by the National Natural Science Foundation of China under Grant 62007013. KRM was funded by the German Ministry for Education and Research as BIFOLD – Berlin Institute for the Foundations of Learning and Data (ref. 01IS18025A and ref. 01IS18037A), and the German Research Foundation (DFG) as Math+: Berlin Mathematics Research Center (EXC 2046/1, project-ID: 390685689), Institute of Information & Communications Technology Planning & Evaluation (IITP) grants funded by the Korea Government (No. 2019-0-00079, Artificial Intelligence Graduate School Program, Korea University). SSN was funded in part by National Institutes of Health grants (R01DC004855, R01EB022717, R01DC176960, R01DC010145, R01NS100440, R01AG062196, and R01DC013979), University of California MRPI MRP-17–454755, the US Department of Defense grant (W81XWH-13-1-0494).

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
