# Supplementary Material: Efficient hierarchical Bayesian inference for spatio-temporal regression models in neuroimaging

## Summary of the proposed algorithms and derived update rules

We proposed two algorithms in the main paper, namely full and thin Dugh, which are summarized in Algorithm 1 and Algorithm 2, respectively.

---

**Algorithm 1:** Full Dugh

---

**Input :** The lead field matrix $\mathbf{L} \in \mathbb{R}^{M \times N}$ and $G$ trials of measurement vectors $\{\mathbf{Y}_g\}_{g=1}^{G}$, where $\mathbf{Y}_g \in \mathbb{R}^{M \times T}$.

**Result:** Estimates of the source and noise variances $\mathbf{h} = [\gamma_1, \dots, \gamma_N, \sigma_1^2, \dots, \sigma_M^2]^\top$, the temporal covariance $\mathbf{B}$, and the posterior mean $\{\bar{\mathbf{x}}_g\}_{g=1}^{G}$ and covariance $\mathbf{\Sigma}_\mathbf{x}$ of the sources.

1 Choose a random initial value for $\mathbf{B}$ as well as $\mathbf{h} = [\gamma_1, \dots, \gamma_N, \sigma_1^2, \dots, \sigma_M^2]^\top$, and construct $\mathbf{H} = \mathrm{diag}(\mathbf{h})$ and $\mathbf{\Gamma} = \mathrm{diag}([\gamma_1, \dots, \gamma_N]^\top)$.

2 Construct the augmented lead field $\mathbf{\Phi} = [\mathbf{L}, \mathbf{I}_M]$.

3 Calculate the lead field $\mathbf{D} = \mathbf{L} \otimes \mathbf{I}_T$ for vectorized sources.

4 Calculate the prior spatio-temporal covariance for the sources as $\mathbf{\Sigma}_0 = \mathbf{\Gamma} \otimes \mathbf{B}$.

5 Calculate the spatial statistical covariance $\mathbf{\Sigma}_\mathbf{y} = \mathbf{\Phi} \mathbf{H} \mathbf{\Phi}^\top$.

6 Calculate the spatio-temporal statistical covariance $\tilde{\mathbf{\Sigma}}_\mathbf{y} = \mathbf{\Sigma}_\mathbf{y} \otimes \mathbf{B}$.

7 Initialize $k \leftarrow 1$

**repeat**

8    Calculate the posterior mean as $\bar{\mathbf{x}}_g = \mathbf{\Sigma}_0 \mathbf{D}^\top \tilde{\mathbf{\Sigma}}_\mathbf{y}^{-1} \mathbf{y}_g$, for $g = 1, \dots, G$, where $\mathbf{y}_g = \mathrm{vec}\left(\mathbf{Y}_g^\top\right) \in \mathbb{R}^{MT \times 1}$.

9    Calculate $\mathbf{M}_{\mathrm{time}}^k$ based on Eq. (6), and update $\mathbf{B}$ based on Eq. (7) according to the Riemannian update on the manifold of P.D. matrices.

10    Calculate $\mathbf{M}_{\mathrm{SN}}^k$ based on Eq. (9), and update $\mathbf{H}$ based on Eq. (10).

11    $k \leftarrow k + 1$

**until** *stopping condition is satisfied:* $\left\|\bar{\mathbf{x}}^{k+1} - \bar{\mathbf{x}}^k\right\|_2^2 \leq \epsilon$ *or* $k = k_{max}$;

12 Calculate the posterior covariance as $\mathbf{\Sigma}_\mathbf{x} = \mathbf{\Sigma}_0 - \mathbf{\Sigma}_0 \mathbf{D}^\top \tilde{\mathbf{\Sigma}}_\mathbf{y}^{-1} \mathbf{D} \mathbf{\Sigma}_0$.

---

**Algorithm 2:** Thin Dugh

---

**Input :** The lead field matrix $\mathbf{L} \in \mathbb{R}^{M \times N}$, and $G$ trials of measurement vectors $\{\mathbf{Y}_g\}_{g=1}^G$, where $\mathbf{Y}_g \in \mathbb{R}^{M \times T}$.

**Result:** Estimates of the source and noise variances $\mathbf{h} = [\gamma_1, \ldots, \gamma_N, \sigma_1^2, \ldots, \sigma_M^2]^\top$, the temporal covariance $\mathbf{B}$, and the posterior mean $\{\bar{\mathbf{x}}_g\}_{g=1}^G$.

1 Choose a random initial value for $\mathbf{p}$ as well as $\mathbf{h}$, and construct $\mathbf{H} = \operatorname{diag}(\mathbf{h})$ and $\mathbf{P} = \operatorname{diag}(\mathbf{p})$.

2 Construct $\mathbf{B} = \mathbf{Q}\mathbf{P}\mathbf{Q}^H$, where $\mathbf{Q} = [\mathbf{I}_M, \mathbf{0}]\mathbf{F}_L$ with $L = 2T + 1$ and $\mathbf{F}_L$ as DFT.

3 Construct the augmented lead field $\mathbf{\Phi} := [\mathbf{L}, \mathbf{I}_M]$.

4 Calculate the lead field $\mathbf{D} = \mathbf{L} \otimes \mathbf{I}_T$ for vectorized sources.

5 Calculate the statistical covariance $\mathbf{\Sigma}_{\mathbf{y}} = \mathbf{\Phi}\mathbf{H}\mathbf{\Phi}^\top$.

6 Calculate the statistical covariance $\mathbf{\Sigma}_{\mathbf{y}} = \mathbf{\Phi}\mathbf{H}\mathbf{\Phi}^\top$.

7 Calculate the spatio-temporal statistical covariance $\tilde{\mathbf{\Sigma}}_{\mathbf{y}} = \mathbf{\Sigma}_{\mathbf{y}} \otimes \mathbf{B}$.

8 Initialize $k \leftarrow 1$

**repeat**

9    Calculate the posterior mean efficiently based on Eq. (19) as
$\bar{\mathbf{x}}_g = \operatorname{tr}\left(\mathbf{Q}\mathbf{P}\left(\mathbf{\Pi} \odot \mathbf{Q}^H \mathbf{Y}_g^\top \mathbf{U}_{\mathbf{x}}\right)\left(\mathbf{U}_{\mathbf{x}}^\top \mathbf{L}\mathbf{\Gamma}^\top\right)\right)$, where $\mathbf{L}\mathbf{\Gamma}\mathbf{L}^\top = \mathbf{U}_{\mathbf{x}}\mathbf{D}_{\mathbf{x}}\mathbf{U}_{\mathbf{x}}^\top$ and
$[\mathbf{\Pi}]_{l,m} = \frac{1}{\sigma_m^2 + p_l d_m}$ for $l = 1, \ldots, L$ and $m = 1, \ldots, M$.

10    Calculate $\mathbf{M}_{\text{time}}^k$ based on Eq. (6), and update $\mathbf{B}$ based on Eq. (16) according to Riemannian update for Toeplitz matrices using circulant embedding.

11    Calculate $\mathbf{M}_{\text{SN}}^k$ based on Eq. (9), and update $\mathbf{H}$ based on Eq. (10).

12    $k \leftarrow k + 1$

**until** *stopping condition is satisfied:* $\left\|\bar{\mathbf{x}}^{k+1} - \bar{\mathbf{x}}^k\right\|_2^2 \leq \epsilon$ *or* $k = k_{max}$;

13 Calculate the posterior covariance as $\mathbf{\Sigma}_{\mathbf{x}} = \mathbf{\Sigma}_0 - \mathbf{\Sigma}_0 \mathbf{D}^\top \tilde{\mathbf{\Sigma}}_{\mathbf{y}}^{-1} \mathbf{D} \mathbf{\Sigma}_0$.

---

# A Derivation of Type-II Bayesian cost function for full-structural spatio-temporal models

In this section, we provide a detailed derivation of Type-II Bayesian learning for full-structural spatio-temporal models. To this end, we first briefly explain the *multiple measurement vector* (MMV) model and then formulate Type-II Bayesian learning with full-structural spatio-temporal covariance structure for this setting. Note that, to simplify the problem, we first present the derivations of the MMV model only for a single trial. We later extend this simplified setting to the multi-trials case.

## A.1 Multiple measurement vector (MMV) model

In M/EEG brain source imaging, a sequence of measurement vectors are often available. Thus, the following *multiple measurement vector (MMV)* model can be formulated1:

$$\mathbf{Y} = \mathbf{L}\mathbf{X} + \mathbf{E} \, ,$$

where $\mathbf{Y} = [\mathbf{y}(1), \ldots, \mathbf{y}(T)] \in \mathbb{R}^{M \times T}$ consists of $T$ measurement vectors for a sequence of $T$ time samples. $\mathbf{X} = [\mathbf{x}(1), \ldots, \mathbf{x}(T)] \in \mathbb{R}^{N \times T}$ is the desired solution matrix (the amplitude of $N$ brain sources during $T$ time samples in our setting), and $\mathbf{E}$ is an unknown noise matrix. A key assumption in the MMV model is that the support (i.e., the indices of the nonzero entries) of every column in $\mathbf{X}$ is identical (referred to as the *common sparsity assumption* in the literature). The number of nonzero rows in $\mathbf{X}$ needs to be below a threshold to ensure unique and global solution. This implies that $\mathbf{X}$ has only a small number of non-zero rows. It has been shown that the recovery of the support can be greatly improved by increasing the number of measurements [50–52].

## A.2 Type-II Bayesian cost function for full-structural spatio-temporal models

To exploit temporal correlations between measurements, we first assume that the voxels are mutually independent. Given the column vector $\boldsymbol{\gamma} = [\gamma_1, \ldots, \gamma_N]^\top$ and a Gaussian probability density for each brain source, the prior distribution with time correlation is modeled as follows:

$$p(\mathbf{X}_i | \gamma_i, \mathbf{B}) \sim \mathcal{N}(0, \gamma_i \mathbf{B}), \qquad i = 1, \ldots, N \, . \tag{20}$$

$\mathbf{X}_i$ denotes the $i$-th row of source matrix $\mathbf{X}$ and models the probability distribution of the $i$-th brain source. Note that $\gamma_i$ is a non-negative hyper-parameter that controls the row sparsity of $\mathbf{X}$; i.e., the values of source $\mathbf{X}_i$ become all zero if $\gamma_i = 0$. Finally, $\mathbf{B}$ is a positive definite matrix that captures the time correlation structure, which is assumed to be shared across all sources. The goal is to obtain the prior distribution of sources, $p(\mathbf{X}|\boldsymbol{\gamma}, \mathbf{B})$, by estimating the hyper-parameters, $\{\boldsymbol{\gamma}, \mathbf{B}\}$. Next, we reformulate the joint MMV model of all sources using vectorization of matrices and Kronecker product operations:

$$\mathbf{y} = \mathbf{D}\mathbf{x} + \mathbf{e} \,,$$

where $\mathbf{y} = \mathrm{vec}\left(\mathbf{Y}^{\top}\right) \in \mathbb{R}^{MT \times 1}$, $\mathbf{x} = \mathrm{vec}\left(\mathbf{X}^{\top}\right) \in \mathbb{R}^{NT \times 1}$, $\mathbf{e} = \mathrm{vec}\left(\mathbf{E}^{\top}\right) \in \mathbb{R}^{MT \times 1}$ and $\mathbf{D}=\mathbf{L} \otimes \mathbf{I}_T$.

The prior distribution of $\mathbf{x}$ is given as

$$p(\mathbf{x}|\boldsymbol{\gamma}, \mathbf{B}) \sim \mathcal{N}(0, \boldsymbol{\Sigma}_0)$$

where $\boldsymbol{\Sigma}_0$ is defined as

$$\boldsymbol{\Sigma}_0 = \begin{bmatrix} \gamma_1 \mathbf{B} & & \\ & \ddots & \\ & & \gamma_N \mathbf{B} \end{bmatrix} = \boldsymbol{\Gamma} \otimes \mathbf{B} \,,$$

in which $\boldsymbol{\Gamma} = \mathrm{diag}(\boldsymbol{\gamma}) = \mathrm{diag}(\gamma_1, \ldots, \gamma_N)$.

Similarly, we may assume zero-mean Gaussian noise with covariance $\boldsymbol{\Sigma}_{\mathbf{e}} = \boldsymbol{\Lambda} \otimes \boldsymbol{\Upsilon}$, where $\mathbf{e} \sim \mathcal{N}(0, \boldsymbol{\Sigma}_{\mathbf{e}})$, and $\boldsymbol{\Lambda}$ and $\boldsymbol{\Upsilon}$ denote the spatial and temporal noise covariance matrices, respectively. Here, we use the same prior for the temporal structure of noise and sources, i.e., $\boldsymbol{\Upsilon} = \mathbf{B}$.

The parameters of the spatio-temporal Type-II model are the unknown source, noise and temporal covariance matrices, i.e., $\boldsymbol{\Gamma}$, $\boldsymbol{\Lambda}$, and $\mathbf{B}$. The unknown parameters $\boldsymbol{\Gamma}$, $\boldsymbol{\Lambda}$, and $\mathbf{B}$ are optimized based on the current estimates of the source, noise and temporal covariances in an alternating iterative process. Given initial estimates of $\boldsymbol{\Gamma}$, $\boldsymbol{\Lambda}$, and $\mathbf{B}$, the posterior distribution of the sources is a Gaussian of the form $p(\mathbf{x}|\mathbf{y}, \boldsymbol{\Gamma}, \boldsymbol{\Lambda}, \mathbf{B}) \sim \mathcal{N}(\bar{\mathbf{x}}, \boldsymbol{\Sigma}_{\mathbf{x}})$, whose mean and covariance are obtained as follows:

$$\bar{\mathbf{x}} = \boldsymbol{\Sigma}_0 \mathbf{D}^{\top} (\boldsymbol{\Lambda} \otimes \mathbf{B} + \mathbf{D}\boldsymbol{\Sigma}_0 \mathbf{D}^{\top})^{-1} \mathbf{y} = \boldsymbol{\Sigma}_0 \mathbf{D}^{\top} \tilde{\boldsymbol{\Sigma}}_{\mathbf{y}}^{-1} \mathbf{y} \,, \tag{21}$$

$$\boldsymbol{\Sigma}_{\mathbf{x}} = \boldsymbol{\Sigma}_0 - \boldsymbol{\Sigma}_0 \mathbf{D}^{\top} \tilde{\boldsymbol{\Sigma}}_{\mathbf{y}}^{-1} \mathbf{D}\boldsymbol{\Sigma}_0 \,, \tag{22}$$

where $\boldsymbol{\Sigma}_{\mathbf{y}} = \mathbf{L}\boldsymbol{\Gamma}\mathbf{L}^{\top} + \boldsymbol{\Lambda}$, and where $\tilde{\boldsymbol{\Sigma}}_{\mathbf{y}} = \boldsymbol{\Sigma}_{\mathbf{y}} \otimes \mathbf{B}$ denotes the spatio-temporal variant of statistical model covariance matrix. The estimated posterior parameters $\bar{\mathbf{x}}$ and $\boldsymbol{\Sigma}_{\mathbf{x}}$ are then in turn used to update $\boldsymbol{\Gamma}$, $\boldsymbol{\Lambda}$, and $\mathbf{B}$ as the minimizers of the negative log of the marginal likelihood $p(\mathbf{Y}|\boldsymbol{\Gamma}, \boldsymbol{\Lambda}, \mathbf{B})$, which is given by

$$\mathcal{L}_{\mathrm{kron}}(\boldsymbol{\Gamma}, \boldsymbol{\Lambda}, \mathbf{B}) = \log|\tilde{\boldsymbol{\Sigma}}_{\mathbf{y}}| + \mathrm{tr}\left(\mathbf{y}^{\top} \tilde{\boldsymbol{\Sigma}}_{\mathbf{y}}^{-1} \mathbf{y}\right) \,. \tag{23}$$

Using the same temporal covariance prior for noise and sources, i.e., $\boldsymbol{\Upsilon} = \mathbf{B}$, the statistical model covariance matrix, $\tilde{\boldsymbol{\Sigma}}_{\mathbf{y}}$, can be written as:

$$\tilde{\boldsymbol{\Sigma}}_{\mathbf{y}} = \boldsymbol{\Lambda} \otimes \boldsymbol{\Upsilon} + (\mathbf{D}\boldsymbol{\Sigma}_0 \mathbf{D}^{\top}) = \boldsymbol{\Lambda} \otimes \boldsymbol{\Upsilon} + \left(\left(\mathbf{L} \otimes \mathbf{I}^{\top}\right)(\boldsymbol{\Gamma} \otimes \mathbf{B})\left(\mathbf{L} \otimes \mathbf{I}^{\top}\right)^{\top}\right)$$

$$= \boldsymbol{\Lambda} \otimes \boldsymbol{\Upsilon} + \left(\mathbf{L}\boldsymbol{\Gamma}\mathbf{L}^{\top} \otimes \mathbf{B}\right) \overset{(\boldsymbol{\Upsilon}=\mathbf{B})}{=} \left(\boldsymbol{\Lambda} + \mathbf{L}\boldsymbol{\Gamma}\mathbf{L}^{\top}\right) \otimes \mathbf{B}$$

$$= \boldsymbol{\Sigma}_{\mathbf{y}} \otimes \mathbf{B} \,, \,. \tag{24}$$

which leads to the following spatio-temporal Type-II Bayesian learning cost function:

$$\mathcal{L}_{\mathrm{kron}}(\boldsymbol{\Gamma}, \boldsymbol{\Lambda}, \mathbf{B}) = \log|\tilde{\boldsymbol{\Sigma}}_{\mathbf{y}}| + \mathrm{tr}\left(\mathbf{y}^{\top}\tilde{\boldsymbol{\Sigma}}_{\mathbf{y}}^{-1}\mathbf{y}\right)$$

$$= \log|\boldsymbol{\Sigma}_{\mathbf{y}} \otimes \mathbf{B}| + \mathrm{tr}\left(\mathbf{y}^{\top}\left(\boldsymbol{\Sigma}_{\mathbf{y}} \otimes \mathbf{B}\right)^{-1}\mathbf{y}\right)$$

$$= \log\left(|\boldsymbol{\Sigma}_{\mathbf{y}}|^{T}|\mathbf{B}|^{M}\right) + \mathrm{tr}\left(\mathbf{y}^{\top}\left(\boldsymbol{\Sigma}_{\mathbf{y}} \otimes \mathbf{B}\right)^{-1}\mathbf{y}\right) \,. \tag{25}$$

Here, we assume the presence of $G$ sample blocks $\mathbf{Y}_g \in \mathbb{R}^{M \times T}$, for $g = 1, \ldots, G$. These block samples can be obtained by segmenting a time series into smaller parts that are assumed to independent

and identically distributed. These blocks may represent epochs, trials or experimentl tasks depending on the applications. $\mathcal{L}_{\text{kron}}(\boldsymbol{\Gamma}, \boldsymbol{\Lambda}, \mathbf{B})$ can then be reformulated as

$$\mathcal{L}_{\text{kron}}(\boldsymbol{\Gamma}, \boldsymbol{\Lambda}, \mathbf{B}) = T \log |\boldsymbol{\Sigma}_{\mathbf{y}}| + M \log |\mathbf{B}| + \frac{1}{G} \sum_{g=1}^{G} \text{tr}(\boldsymbol{\Sigma}_{\mathbf{y}}^{-1} \mathbf{Y}_g \mathbf{B}^{-1} \mathbf{Y}_g^{\top}) \tag{26}$$

by applying the following matrix equality to Eq. (25):

$$\text{tr}(\mathbf{A}^{\top} \mathbf{B} \mathbf{C} \mathbf{D}^{\top}) = \text{vec}(\mathbf{A})^{\top} (\mathbf{D} \otimes \mathbf{B}) \text{vec}(\mathbf{C}) .$$

# B Proof of Theorem 1

*Proof.* We start by recalling $\mathcal{L}_{\text{kron}}(\boldsymbol{\Gamma}, \boldsymbol{\Lambda}, \mathbf{B})$ in Eq. (4):

$$\mathcal{L}_{\text{kron}}(\boldsymbol{\Gamma}, \boldsymbol{\Lambda}, \mathbf{B}) = T \log |\boldsymbol{\Sigma}_{\mathbf{y}}| + M \log |\mathbf{B}| + \frac{1}{G} \sum_{g=1}^{G} \text{tr}(\boldsymbol{\Sigma}_{\mathbf{y}}^{-1} \mathbf{Y}_g \mathbf{B}^{-1} \mathbf{Y}_g^{\top}).$$

Let $\boldsymbol{\Sigma}_{\mathbf{y}}^{k}, \boldsymbol{\Gamma}^{k}$, and $\boldsymbol{\Lambda}^{k}$ be the values of statistical model covariance and the source and noise covariances at the $k$-th iteration, respectively. By ignoring terms that do not depend on $\mathbf{B}$, $\mathcal{L}_{\text{kron}}(\boldsymbol{\Gamma}, \boldsymbol{\Lambda}, \mathbf{B})$ can be written as follows:

$$\begin{aligned}
\mathcal{L}_{\text{kron}}^{\text{time}}(\boldsymbol{\Gamma}^{k}, \boldsymbol{\Lambda}^{k}, \mathbf{B}) &= M \log |\mathbf{B}| + \frac{1}{G} \sum_{g=1}^{G} \text{tr}\left( \left(\boldsymbol{\Sigma}_{\mathbf{y}}^{k}\right)^{-1} \mathbf{Y}_g \mathbf{B}^{-1} \mathbf{Y}_g^{\top} \right) \\
&= \log |\mathbf{B}| + \frac{1}{MG} \sum_{g=1}^{G} \text{tr}\left( \left(\boldsymbol{\Sigma}_{\mathbf{y}}^{k}\right)^{-1} \mathbf{Y}_g \mathbf{B}^{-1} \mathbf{Y}_g^{\top} \right) \\
&= \log |\mathbf{B}| + \text{tr}\left( \mathbf{B}^{-1} \frac{1}{MG} \sum_{g=1}^{G} \mathbf{Y}_g^{\top} \left(\boldsymbol{\Sigma}_{\mathbf{y}}^{k}\right)^{-1} \mathbf{Y}_g \right) \\
&= \log |\mathbf{B}| + \text{tr}\left( \mathbf{B}^{-1} \mathbf{M}_{\text{time}}^{k} \right) ,
\end{aligned} \tag{27}$$

where $\mathbf{M}_{\text{time}}^{k} := \frac{1}{MG} \sum_{g=1}^{G} \mathbf{Y}_g^{\top} \left(\boldsymbol{\Sigma}_{\mathbf{y}}^{k}\right)^{-1} \mathbf{Y}_g$.

By virtue of the concavity of the log-determinant function and its first order Taylor expansion around $\mathbf{B}^{k}$, the following inequality holds:

$$\begin{aligned}
\mathcal{L}_{\text{kron}}^{\text{time}}(\boldsymbol{\Gamma}^{k}, \boldsymbol{\Lambda}^{k}, \mathbf{B}) &= \log |\mathbf{B}| + \text{tr}\left( \mathbf{B}^{-1} \mathbf{M}_{\text{time}}^{k} \right) \\
&\leq \log \left| \mathbf{B}^{k} \right| + \text{tr}\left( \left(\mathbf{B}^{k}\right)^{-1} \left( \mathbf{B} - \mathbf{B}^{k} \right) \right) + \text{tr}\left( \mathbf{B}^{-1} \mathbf{M}_{\text{time}}^{k} \right) \\
&= \log \left| \mathbf{B}^{k} \right| + \text{tr}\left( \left(\mathbf{B}^{k}\right)^{-1} \mathbf{B} \right) - \text{tr}\left( \left(\mathbf{B}^{k}\right)^{-1} \mathbf{B}^{k} \right) + \text{tr}\left( \mathbf{B}^{-1} \mathbf{M}_{\text{time}}^{k} \right) \\
&= \text{tr}\left( (\mathbf{B}^{k})^{-1} \mathbf{B} \right) + \text{tr}(\mathbf{B}^{-1} \mathbf{M}_{\text{time}}^{k}) + \text{const} \\
&= \mathcal{L}_{\text{conv}}^{\text{time}}(\boldsymbol{\Gamma}^{k}, \boldsymbol{\Lambda}^{k}, \mathbf{B}) + \text{const} .
\end{aligned} \tag{28}$$

Note that constant values in (28) do not depend on $\mathbf{B}$; hence, they can be ignored in the optimization procedure. Hence, we have shown that minimizing Eq. (4) with respect to $\mathbf{B}$ is equivalent to minimizing $\mathcal{L}_{\text{conv}}^{\text{time}}(\boldsymbol{\Gamma}^{k}, \boldsymbol{\Lambda}^{k}, \mathbf{B})$, which concludes the proof. $\qquad\square$

# C Proof of Theorem 2

Before presenting the proof, the subsequent definitions and propositions are required:

**Definition 1** (Geodesic path). *Let $\mathcal{M}$ be a Riemannian manifold, i.e., a differentiable manifold whose tangent space is endowed with an inner product that defines local Euclidean structures. Then, a geodesic between two points on $\mathcal{M}$, denoted by $\mathbf{p}_0, \mathbf{p}_1 \in \mathcal{M}$, is defined as the shortest connecting path between those two points along the manifold, $\zeta_l(\mathbf{p}_0, \mathbf{p}_1) \in \mathcal{M}$ for $l \in [0, 1]$, where $l = 0$ and $l = 1$ defines the starting and end points of the path, respectively.*

In the current context, $\zeta_l(\mathbf{p}_0, \mathbf{p}_1)$ defines a geodesic curve on the P.D. manifold joining two P.D. matrices, $\mathbf{P}_0, \mathbf{P}_1 > 0$. The specific pair of matrices we will deal with is $\{\mathbf{B}^k, \mathbf{M}^k_{\text{time}}\}$.

**Definition 2** (Geodesic on the P.D. manifold). *Geodesics on the manifold of P.D. matrices can be shown to form a cone within the embedding space. We denote this manifold by $\mathcal{S}_{++}$. Assume two P.D. matrices $\mathbf{P}_0, \mathbf{P}_1 \in \mathcal{S}_{++}$. Then, for $l \in [0, 1]$, the geodesic curve joining $\mathbf{P}_0$ to $\mathbf{P}_1$ is defined as [53, Chapter. 6]:*

$$\xi_l(\mathbf{P}_0, \mathbf{P}_1) = (\mathbf{P}_0)^{\frac{1}{2}} \left( (\mathbf{P}_0)^{-1/2} \mathbf{P}_1 (\mathbf{P}_0)^{-1/2} \right)^l (\mathbf{P}_0)^{\frac{1}{2}} \quad l \in [0, 1] . \tag{29}$$

Note that $\mathbf{P}_0$ and $\mathbf{P}_1$ are obtained as the starting and end points of the geodesic path by choosing $l = 0$ and $l = 1$, respectively. The midpoint of the geodesic, obtained by setting $l = \frac{1}{2}$, is called the *geometric mean*. Note that, according to Definition 2, the following equality holds :

$$\xi_l(\mathbf{B}_0, \mathbf{B}_1)^{-1} = \left( (\mathbf{B}_0)^{1/2} \left( (\mathbf{B}_0)^{-1/2} \mathbf{B}_1 (\mathbf{B}_0)^{-1/2} \right)^l (\mathbf{B}_0)^{1/2} \right)^{-1}$$

$$= \left( (\mathbf{B}_0)^{-1/2} \left( (\mathbf{B}_0)^{1/2} (\mathbf{B}_1)^{-1} (\mathbf{B}_0)^{1/2} \right)^l (\mathbf{B}_0)^{-1/2} \right) = \xi_l(\mathbf{B}_0^{-1}, \mathbf{B}_1^{-1}) . \tag{30}$$

**Definition 3** (Geodesic convexity). *Let $\mathbf{p}_0$ and $\mathbf{p}_1$ be two arbitrary points on a subset $\mathcal{A}$ of a Riemannian manifold $\mathcal{M}$. Then, a real-valued function $f : \mathcal{A} \to \mathbb{R}$ with domain $\mathcal{A} \subset \mathcal{M}$ is called geodesic convex (g-convex) if the following relation holds:*

$$f \left( \zeta_l(\mathbf{p}_0, \mathbf{p}_1) \right) \leq l f(\mathbf{p}_0) + (1 - l) f(\mathbf{p}_1) , \tag{31}$$

*where $l \in [0, 1]$ and $\zeta_l(\mathbf{p}_0, \mathbf{p}_1)$ denotes the geodesic path connecting two points $\mathbf{p}_0$ and $\mathbf{p}_1$ as defined in Definition 1. Thus, in analogy to classical convexity, the function $f$ is g-convex if every geodesic $\zeta(\mathbf{p}_0, \mathbf{p}_1)$ of $\mathcal{M}$ between $\mathbf{p}_0, \mathbf{p}_1 \in \mathcal{A}$, lies in the g-convex set $\mathcal{A}$. Note that the set $\mathcal{A} \subset \mathcal{M}$ is called g-convex, if any geodesics joining an arbitrary pair of points lies completely in $\mathcal{A}$.*

**Remark 2.** *Note that g-convexity is a generalization of classical (linear) convexity to non-Euclidean (non-linear) geometry and metric spaces. Therefore, it is straightforward to show that all convex functions in Euclidean geometry are also g-convex, where the geodesics between pairs of matrices are simply line segments:*

$$\zeta_l(\mathbf{p}_0, \mathbf{p}_1) = l \mathbf{p}_0 + (1 - l) \mathbf{p}_1 . \tag{32}$$

For the sake of brevity, we omit a detailed theoretical introduction of g-convexity, and only borrow a result from [54]. Interested readers are referred to [55, Chapter 1] for a gentle introduction to this topic, and [56, Chapter. 2]; [57–64] for more in-depth technical details. Now we are ready to state the proof, which parallels the one provided in Zadeh et al. [54, Theorem. 3].

*Proof.* We proceed in two steps. First, we consider P.D. manifolds and express (31) in terms of geodesic paths and functions that lie on this particular space. We then show that $\mathcal{L}^{\text{time}}_{\text{conv}}(\boldsymbol{\Gamma}^k, \boldsymbol{\Lambda}^k, \mathbf{B})$ is strictly g-convex on this specific domain. In the second step, we then derive the update rule proposed in (7).

### C.1 Part I: G-convexity of the majorizing cost function

We consider geodesics along the P.D. manifold by setting $\zeta_l(\mathbf{p}_0, \mathbf{p}_1)$ to $\xi_l(\mathbf{B}_0, \mathbf{B}_1)$ as presented in Definition 2, and define $f(.)$ to be $f(\mathbf{B}) = \text{tr} \left( \left( \mathbf{B}^k \right)^{-1} \mathbf{B} \right) + \text{tr}(\mathbf{M}^k_{\text{time}} \mathbf{B}^{-1})$, representing the cost function $\mathcal{L}^{\text{time}}_{\text{conv}}(\boldsymbol{\Gamma}^k, \boldsymbol{\Lambda}^k, \mathbf{B})$.

We now show that $f(\mathbf{B})$ is strictly g-convex on this specific domain. For continuous functions as considered in this paper, fulfilling (31) for $f(\mathbf{B})$ and $\xi_l(\mathbf{B}_0, \mathbf{B}_1)$ with $l = 1/2$ is sufficient for strict g-convexity according to *mid-point convexity* [65]:

$$\text{tr} \left( \left( \mathbf{B}^k \right)^{-1} \xi_{1/2}(\mathbf{B}_0, \mathbf{B}_1) \right) + \text{tr} \left( \mathbf{M}^k_{\text{time}} \xi_{1/2}(\mathbf{B}_0, \mathbf{B}_1)^{-1} \right)$$

$$< \frac{1}{2} \text{tr} \left( \left( \mathbf{B}^k \right)^{-1} \mathbf{B}_0 \right) + \frac{1}{2} \text{tr} \left( \mathbf{M}^k_{\text{time}} \mathbf{B}_0^{-1} \right)$$

$$+ \frac{1}{2} \text{tr} \left( \left( \mathbf{B}^k \right)^{-1} \mathbf{B}_1 \right) + \frac{1}{2} \text{tr} \left( \mathbf{M}^k_{\text{time}} \mathbf{B}_1^{-1} \right) . \tag{33}$$

Given $\left(\mathbf{B}^k\right)^{-1} \in \mathcal{S}_{++}$, i.e., $\left(\mathbf{B}^k\right)^{-1} > 0$ and the operator inequality [53, Chapter. 4]

$$\xi_{1/2}(\mathbf{B}_0, \mathbf{B}_1) \prec \frac{1}{2}\mathbf{B}_0 + \frac{1}{2}\mathbf{B}_1 \;, \tag{34}$$

we have:

$$\mathrm{tr}\left(\left(\mathbf{B}^k\right)^{-1}\xi_{1/2}(\mathbf{B}_0, \mathbf{B}_1)\right) < \frac{1}{2}\mathrm{tr}\left(\left(\mathbf{B}^k\right)^{-1}\mathbf{B}_0\right) + \frac{1}{2}\mathrm{tr}\left(\left(\mathbf{B}^k\right)^{-1}\mathbf{B}_1\right) \;, \tag{35}$$

which is derived by multiplying both sides of Eq. (34) with $\left(\mathbf{B}^k\right)^{-1}$ followed by taking the trace on both sides.

Similarly, we can write the operator inequality for $\{\mathbf{B}_0^{-1}, \mathbf{B}_1^{-1}\}$ using Eq. (30) as:

$$\xi_{1/2}(\mathbf{B}_0, \mathbf{B}_1)^{-1} = \xi_{1/2}(\mathbf{B}_0^{-1}, \mathbf{B}_1^{-1}) \prec \frac{1}{2}\mathbf{B}_0^{-1} + \frac{1}{2}\mathbf{B}_1^{-1} \;. \tag{36}$$

Multiplying both sides of Eq. (36) by $\mathbf{M}_{\mathrm{time}}^k \in \mathcal{S}_{++}$ and applying the trace operator on both sides leads to:

$$\mathrm{tr}\left(\mathbf{M}_{\mathrm{time}}^k\xi_{1/2}(\mathbf{B}_0, \mathbf{B}_1)^{-1}\right) < \frac{1}{2}\mathrm{tr}\left(\mathbf{M}_{\mathrm{time}}^k\mathbf{B}_0^{-1}\right) + \frac{1}{2}\mathrm{tr}\left(\mathbf{M}_{\mathrm{time}}^k\mathbf{B}_1^{-1}\right) \;. \tag{37}$$

Summing up (35) and (37) proves inequality (33) and concludes the first part of the proof.

### C.2   Part II: Derivation of the update rule in Eq. (7)

We now present the second part of the proof by deriving the update rule in Eq. (7). Since the cost function $\mathcal{L}_{\mathrm{conv}}^{\mathrm{time}}(\mathbf{\Gamma}^k, \mathbf{\Lambda}^k, \mathbf{B})$ is strictly g-convex, its optimal solution in the $k$-th iteration is unique. More concretely, the optimum can be analytically derived by taking the derivative of Eq. (7) and setting the result to zero as follows:

$$\nabla\mathcal{L}_{\mathrm{conv}}^{\mathrm{time}}(\mathbf{\Gamma}^k, \mathbf{\Lambda}^k, \mathbf{B}) = \left(\mathbf{B}^k\right)^{-1} - \mathbf{B}^{-1}\mathbf{M}_{\mathrm{time}}^k\mathbf{B}^{-1} = 0 \;, \tag{38}$$

which results in

$$\mathbf{B}\left(\mathbf{B}^k\right)^{-1}\mathbf{B} = \mathbf{M}_{\mathrm{time}}^k \;. \tag{39}$$

This solution is known as the *Riccati equation* and is the geometric mean between $\mathbf{B}^k$ and $\mathbf{M}_{\mathrm{time}}^k$ [61, 66]:

$$\mathbf{B}^{k+1} \leftarrow (\mathbf{B}^k)^{\frac{1}{2}}\left((\mathbf{B}^k)^{-1/2}\mathbf{M}_{\mathrm{time}}^k(\mathbf{B}^k)^{-1/2}\right)^{\frac{1}{2}}(\mathbf{B}^k)^{\frac{1}{2}} \;.$$

Deriving the update rule in Eq. (7) concludes the second part of the proof of Theorem 2. $\square$

## D   Proof of Theorem 3

*Proof.* Analogous to the proof of Theorem 1 in Appendix B, we start by recalling $\mathcal{L}_{\mathrm{kron}}(\mathbf{\Gamma}, \mathbf{\Lambda}, \mathbf{B})$ in Eq. (4):

$$\mathcal{L}_{\mathrm{kron}}(\mathbf{\Gamma}, \mathbf{\Lambda}, \mathbf{B}) = T\log|\mathbf{\Sigma_y}| + M\log|\mathbf{B}| + \frac{1}{G}\sum_{g=1}^{G}\mathrm{tr}(\mathbf{\Sigma_y}^{-1}\mathbf{Y}_g\mathbf{B}^{-1}\mathbf{Y}_g^\top).$$

Let $\mathbf{B}^k$ be the value of the temporal covariance matrix learned using Eq. (7) at $k$-th iteration. Then, by ignoring the term $M\log|\mathbf{B}^k|$ that is only a function of $\mathbf{B}^k$, $\mathcal{L}_{\mathrm{kron}}(\mathbf{\Gamma}, \mathbf{\Lambda}, \mathbf{B})$ can be written as follows:

$$\begin{aligned}
\mathcal{L}_{\mathrm{kron}}^{\mathrm{space}}(\mathbf{\Gamma}, \mathbf{\Lambda}, \mathbf{B}^k) &= T\log|\mathbf{\Sigma_y}| + \frac{1}{G}\sum_{g=1}^{G}\mathrm{tr}\left(\mathbf{\Sigma_y}^{-1}\mathbf{Y}_g(\mathbf{B}^k)^{-1}\mathbf{Y}_g^\top\right) \\
&= \log|\mathbf{\Sigma_y}| + \frac{1}{TG}\sum_{g=1}^{G}\mathrm{tr}\left(\mathbf{\Sigma_y}^{-1}\mathbf{Y}_g(\mathbf{B}^k)^{-1}\mathbf{Y}_g^\top\right) \\
&= \log|\mathbf{\Sigma_y}| + \mathrm{tr}\left(\mathbf{\Sigma_y}^{-1}\frac{1}{TG}\sum_{g=1}^{G}\mathbf{Y}_g(\mathbf{B}^k)^{-1}\mathbf{Y}_g^\top\right) \\
&= \log|\mathbf{\Sigma_y}| + \mathrm{tr}\left(\mathbf{\Sigma_y}^{-1}\mathbf{M}_{\mathrm{space}}^k\right) \;,
\end{aligned} \tag{40}$$

where $\mathbf{M}^k_{\text{space}} := \frac{1}{TG} \sum_{g=1}^{G} \mathbf{Y}_g (\mathbf{B}^k)^{-1} \mathbf{Y}_g^\top$.

Similar to the argument made in Appendix B, a first order Taylor expansion of the log-determinant function around $\boldsymbol{\Sigma}_\mathbf{y}$ provides the following inequality:

$$\log |\boldsymbol{\Sigma}_\mathbf{y}| \leq \log \left|\boldsymbol{\Sigma}_\mathbf{y}^k\right| + \text{tr}\left(\left(\boldsymbol{\Sigma}_\mathbf{y}^k\right)^{-1} \left(\boldsymbol{\Sigma}_\mathbf{y} - \boldsymbol{\Sigma}_\mathbf{y}^k\right)\right)$$
$$= \log \left|\boldsymbol{\Sigma}_\mathbf{y}^k\right| + \text{tr}\left(\left(\boldsymbol{\Sigma}_\mathbf{y}^k\right)^{-1} \boldsymbol{\Sigma}_\mathbf{y}\right) - \text{tr}\left(\left(\boldsymbol{\Sigma}_\mathbf{y}^k\right)^{-1} \boldsymbol{\Sigma}_\mathbf{y}^k\right)$$
$$= \text{tr}(\boldsymbol{\Phi}^\top (\boldsymbol{\Sigma}_\mathbf{y}^k)^{-1} \boldsymbol{\Phi}\mathbf{H}) + \text{const}, \tag{41}$$

where the last step is derived using the augmented source and noise covariances, $\mathbf{H} := [\boldsymbol{\Gamma}, \mathbf{0}; \mathbf{0}, \boldsymbol{\Lambda}]$, $\boldsymbol{\Phi} := [\mathbf{L}, \mathbf{I}]$ and $\boldsymbol{\Sigma}_\mathbf{y} = \boldsymbol{\Phi}\mathbf{H}\boldsymbol{\Phi}^\top$.

By inserting Eq. (40) into Eq. (41), the first term of Eq. (8), $\text{tr}(\boldsymbol{\Phi}^\top (\boldsymbol{\Sigma}_\mathbf{y}^k)^{-1} \boldsymbol{\Phi}\mathbf{H})$, can be directly inferred:

$$\mathcal{L}^{\text{space}}_{\text{kron}}(\boldsymbol{\Gamma}, \boldsymbol{\Lambda}, \mathbf{B}^k) = \mathcal{L}^{\text{space}}_{\text{kron}}(\mathbf{H}, \mathbf{B}^k) = \log |\boldsymbol{\Sigma}_\mathbf{y}| + \text{tr}\left(\boldsymbol{\Sigma}_\mathbf{y}^{-1} \mathbf{M}^k_{\text{space}}\right)$$
$$\leq \text{tr}(\boldsymbol{\Phi}^\top (\boldsymbol{\Sigma}_\mathbf{y}^k)^{-1} \boldsymbol{\Phi}\mathbf{H}) + \text{tr}\left(\boldsymbol{\Sigma}_\mathbf{y}^{-1} \mathbf{M}^k_{\text{space}}\right) + \text{const}, \tag{42}$$

We further show how the second term in Eq. (8) can be derived. To this end, we construct an upper bound on $\text{tr}\left(\boldsymbol{\Sigma}_\mathbf{y}^{-1} \mathbf{M}^k_{\text{space}}\right)$ using an inequality derived from the Schur complement of $\boldsymbol{\Sigma}_\mathbf{y}$. Before presenting this inequality, the subsequent definition of the Schur complement of matrix $\boldsymbol{\Sigma}_\mathbf{y}$ is required:

**Definition 4.** *For a positive semidefinite (PSD) matrix $\boldsymbol{\Sigma}_\mathbf{y}$, and a partitioning*

$$\mathbf{X} = \begin{bmatrix} \mathbf{D} & \mathbf{G} \\ \mathbf{G}^\top & \mathbf{B} \end{bmatrix}, \tag{43}$$

*its Schur complement is defined as*

$$\mathbf{S} := \mathbf{D} - \mathbf{G}\boldsymbol{\Sigma}_\mathbf{y}^{-1} \mathbf{G}^\top \tag{44}$$

$$\tag{45}$$

*The Schur complement condition states that the matrix $\mathbf{X}$ is PSD, $\mathbf{X} \geq \mathbf{0}$, if and only if the Schur complement of $\boldsymbol{\Sigma}_\mathbf{y}$ is PSD, $\mathbf{S} \geq \mathbf{0}$.*

Now we are ready to construct an upper bound on $\text{tr}\left(\boldsymbol{\Sigma}_\mathbf{y}^{-1} \mathbf{M}^k_{\text{space}}\right)$. To this end, we show that $\text{tr}\left(\boldsymbol{\Sigma}_\mathbf{y}^{-1} \mathbf{M}^k_{\text{space}}\right)$ can be majorized as follows:

$$\text{tr}\left(\boldsymbol{\Sigma}_\mathbf{y}^{-1} \mathbf{M}^k_{\text{space}}\right) \leq \text{tr}(\mathbf{H}^k \boldsymbol{\Phi}^\top (\boldsymbol{\Sigma}_\mathbf{y}^k)^{-1} \mathbf{M}^k_{\text{space}} (\boldsymbol{\Sigma}_\mathbf{y}^k)^{-1} \boldsymbol{\Phi}\mathbf{H}^k \mathbf{H}^{-1}). \tag{46}$$

By defining $\mathbf{V}$ as:

$$\mathbf{V} = \begin{bmatrix} (\boldsymbol{\Sigma}_\mathbf{y}^k)^{-1} \boldsymbol{\Phi}\mathbf{H}^k \mathbf{H}^{\frac{-1}{2}} \\ \boldsymbol{\Phi}\mathbf{H}^{\frac{1}{2}} \end{bmatrix}, \tag{47}$$

the PSD property of $\mathbf{S}$ can be inferred as:

$$\mathbf{S} = \begin{bmatrix} (\boldsymbol{\Sigma}_\mathbf{y}^k)^{-1} \boldsymbol{\Phi}\mathbf{H}^k \mathbf{H}^{-1} \mathbf{H}^k \boldsymbol{\Phi}^\top (\boldsymbol{\Sigma}_\mathbf{y}^k)^{-1} & \mathbf{I} \\ \mathbf{I} & \boldsymbol{\Phi}\mathbf{H}\boldsymbol{\Phi}^\top \end{bmatrix} = \mathbf{V}\mathbf{V}^\top \geq 0. \tag{48}$$

By employing the definition of the Schur complement with $\mathbf{D} = (\boldsymbol{\Sigma}_\mathbf{y}^k)^{-1} \boldsymbol{\Phi}\mathbf{H}^k \mathbf{H}^{-1} \mathbf{H}^k \boldsymbol{\Phi}^\top (\boldsymbol{\Sigma}_\mathbf{y}^k)^{-1}$, $\mathbf{G} = \mathbf{I}$ and $\boldsymbol{\Sigma}_\mathbf{y} = \boldsymbol{\Phi}\mathbf{H}\boldsymbol{\Phi}^\top$, we have:

$$(\boldsymbol{\Sigma}_\mathbf{y}^k)^{-1} \boldsymbol{\Phi}\mathbf{H}^k \mathbf{H}^{-1} \mathbf{H}^k \boldsymbol{\Phi}^\top (\boldsymbol{\Sigma}_\mathbf{y}^k)^{-1} \geq \left(\boldsymbol{\Phi}\mathbf{H}\boldsymbol{\Phi}^\top\right)^{-1}. \tag{49}$$

The inequality in Eq. (46) can be directly inferred by multiplying $\mathbf{M}^k_{\text{space}}$ to both sides of Eq. (49), applying trace operator, and rearranging the arguments in the trace operator:

$$\text{tr}(\mathbf{M}^k_{\text{space}} \boldsymbol{\Sigma}_\mathbf{y}^{-1}] \leq \text{tr}(\mathbf{M}^k_{\text{space}} (\boldsymbol{\Sigma}_\mathbf{y}^k)^{-1} \boldsymbol{\Phi}\mathbf{H}^k \mathbf{H}^{-1} \mathbf{H}^k \boldsymbol{\Phi}^\top (\boldsymbol{\Sigma}_\mathbf{y}^k)^{-1})$$
$$= \text{tr}(\mathbf{H}^k \boldsymbol{\Phi}^\top (\boldsymbol{\Sigma}_\mathbf{y}^k)^{-1} \mathbf{M}^k_{\text{space}} (\boldsymbol{\Sigma}_\mathbf{y}^k)^{-1} \boldsymbol{\Phi}\mathbf{H}^k \mathbf{H}^{-1})$$
$$= \text{tr}\left(\mathbf{M}^k_{\text{SN}} \mathbf{H}^{-1}\right). \tag{50}$$

By inserting Eq. (50) into Eq. (42), we have

$$
\begin{aligned}
\mathcal{L}_{\text{kron}}^{\text{space}}(\boldsymbol{\Gamma}, \boldsymbol{\Lambda}, \mathbf{B}^k) = \mathcal{L}_{\text{kron}}^{\text{space}}(\mathbf{H}, \mathbf{B}^k) &\leq \text{tr}(\boldsymbol{\Phi}^\top (\boldsymbol{\Sigma}_\mathbf{y}^k)^{-1} \boldsymbol{\Phi} \mathbf{H}) + \text{tr}\left(\boldsymbol{\Sigma}_\mathbf{y}^{-1} \mathbf{M}_{\text{space}}^k\right) + \text{const} \\
&\leq \text{tr}(\boldsymbol{\Phi}^\top (\boldsymbol{\Sigma}_\mathbf{y}^k)^{-1} \boldsymbol{\Phi} \mathbf{H}) + \text{tr}\left(\mathbf{M}_{\text{SN}}^k \mathbf{H}^{-1}\right) + \text{const} \\
&= \mathcal{L}_{\text{conv}}^{\text{space}}(\mathbf{H}, \mathbf{B}^k) + \text{const} .
\end{aligned}
\tag{51}
$$

Note that constant values in (51) do not depend on $\mathbf{H}$; hence, they can be ignored in the optimization procedure. We have shown that minimizing Eq. (4) with respect to $\mathbf{H}$ is equivalent to minimizing $\mathcal{L}_{\text{conv}}^{\text{space}}(\mathbf{H}, \mathbf{B}^k)$, which concludes the proof. □

# E  Proof of Theorem 4

*Proof.* We proceed in two steps. First, we show that $\mathcal{L}_{\text{conv}}^{\text{space}}(\mathbf{H}, \mathbf{B}^k)$ is convex in $\mathbf{h}$. Then, we derive the update rule proposed in Eq. (10).

## E.1  Part I: Convexity of the majorizing cost function

We start the proof by constraining $\mathbf{H}$ to the set of diagonal matrices with non-negative entries $\mathcal{S}$, i.e., $\mathcal{S} = \{\mathbf{H} \mid \mathbf{H} = \text{diag}(\mathbf{h}) = \text{diag}([h_1, \ldots, h_{N+M}]^\top), \; h_n \geq 0, \; \text{for } i = 1, \ldots, N + M\}$. We continue by reformulating the constrained optimization with respect to the source covariance matrix,

$$
\mathbf{H}^{k+1} = \underset{\mathbf{H} \in \mathcal{S}, \; \mathbf{B} = \mathbf{B}^k}{\arg\min} \; \text{tr}\left(\boldsymbol{\Phi}^\top \left(\boldsymbol{\Sigma}_\mathbf{y}^k\right)^{-1} \boldsymbol{\Phi} \mathbf{H}\right) + \text{tr}(\mathbf{M}_{\text{SN}}^k \mathbf{H}^{-1}) ,
\tag{52}
$$

as follows:

$$
\mathbf{h}^{k+1} = \underset{\mathbf{h} \geq 0, \; \mathbf{B} = \mathbf{B}^k}{\arg\min} \; \underbrace{\text{diag}\left(\boldsymbol{\Phi}^\top \left(\boldsymbol{\Sigma}_\mathbf{y}^k\right)^{-1} \boldsymbol{\Phi}\right) \mathbf{h} + \text{diag}\left(\mathbf{M}_{\text{SN}}^k\right) \mathbf{h}^{-1}}_{\mathcal{L}_{\text{diag}}^{\text{space}}(\mathbf{h}|\mathbf{h}^k)} ,
\tag{53}
$$

where $\mathbf{h}^{-1} = [h_1^{-1}, \ldots, h_N^{-1}]^\top$ is defined as the element-wise inversion of $\mathbf{h}$. Let $\mathbf{V}^k := \boldsymbol{\Phi}^\top \left(\boldsymbol{\Sigma}_\mathbf{y}^k\right)^{-1} \boldsymbol{\Phi}$. Then, we rewrite $\mathcal{L}_{\text{diag}}^{\text{space}}(\mathbf{h}|\mathbf{h}^k)$ as

$$
\mathcal{L}_{\text{diag}}^{\text{space}}(\mathbf{h}|\mathbf{h}^k) = \text{diag}\left(\mathbf{V}^k\right) \mathbf{h} + \text{diag}\left(\mathbf{M}_{\text{SN}}^k\right) \mathbf{h}^{-1} .
\tag{54}
$$

The convexity of $\mathcal{L}_{\text{diag}}^{\text{space}}(\mathbf{h}|\mathbf{h}^k)$ can be directly inferred from the convexity of $\text{diag}\left[\mathbf{V}^k\right] \mathbf{h}$ and $\text{diag}\left[\mathbf{M}_{\text{SN}}^k\right] \mathbf{h}^{-1}$ with respect to $\mathbf{h}$ [67, Chapter. 3].

## E.2  Part II: Derivation of the update rule in Eq. (10)

We now present the second part of the proof by deriving the update rule in Eq. (10). Since the cost function $\mathcal{L}_{\text{diag}}^{\text{space}}(\mathbf{h}|\mathbf{h}^k)$ is convex, its optimal solution in the $k$-th iteration is unique. Therefore, the optimization with respect to heteroscedastic source and noise variances is carried out by taking the derivative of (53) with respect to $h_i$, for $n = 1, \ldots, M + N$, and setting it to zero:

$$
\begin{aligned}
\frac{\partial}{\partial h_i} &\left(\left[\boldsymbol{\Phi}^\top \left(\boldsymbol{\Sigma}_\mathbf{y}^k\right)^{-1} \boldsymbol{\Phi}\right] h_i + \left[\mathbf{M}_{\text{SN}}^k\right] h_i^{-1}\right) \\
&= \left[\boldsymbol{\Phi}^\top \left(\boldsymbol{\Sigma}_\mathbf{y}^k\right)^{-1} \boldsymbol{\Phi}\right]_{i,i} - \frac{1}{(h_i)^2} \left[\mathbf{M}_{\text{SN}}^k\right]_{i,i} \\
&= 0 \quad \text{for } i = 1, \ldots, N + M ,
\end{aligned}
$$

where $\boldsymbol{\Phi}_i$ denotes the $n$-th column of the augmented lead field matrix. This yields the following update rule:

$$
\mathbf{H}^{k+1} = \text{diag}(\mathbf{h}^{k+1}), \quad \text{where,} \quad h_i^{k+1} \leftarrow \sqrt{\frac{\left[\mathbf{M}_{\text{SN}}^k\right]_{i,i}}{\left[\boldsymbol{\Phi}^\top \left(\boldsymbol{\Sigma}_\mathbf{y}^k\right)^{-1} \boldsymbol{\Phi}\right]_{i,i}}} = \sqrt{\frac{\frac{1}{T}\sum_{t=1}^{T}(\bar{\boldsymbol{\eta}}_n^k(t))^2}{\boldsymbol{\Phi}_n^\top \left(\boldsymbol{\Sigma}_\mathbf{y}^k\right)^{-1} \boldsymbol{\Phi}_i}}
$$
$$
\text{for } i = 1, \ldots, N + M .
\tag{55}
$$

The updates rule in Eq. (10) can be directly inferred by defining $\mathbf{g} := \text{diag}(\mathbf{M}_{\text{SN}}^k)$ and $\mathbf{z} := \text{diag}(\boldsymbol{\Phi}^\top(\boldsymbol{\Sigma}_\mathbf{y}^k)^{-1}\boldsymbol{\Phi})$, which leads to: $g_i^k = \left[\mathbf{M}_{\text{SN}}^k\right]_{i,i}$ and $z_i^k = \left[\boldsymbol{\Phi}^\top \left(\boldsymbol{\Sigma}_\mathbf{y}^k\right)^{-1} \boldsymbol{\Phi}\right]_{i,i}$. This concludes the proof. □

## F Champagne with heteroscedastic noise learning

Interestingly, identical update rules as those proposed in Champagne [33] and heteroscedastic noise learning [68] can be derived for source and noise variances, respectively, by selecting the corresponding indices of matrix $\mathbf{H}$ associated to noise and source covariances.

### F.1 Update rule for source variances

Given $[\mathbf{\Phi}]_{1:M,1:N} = \mathbf{L}$, $[\mathbf{H}]_{1:N,1:N} = \mathbf{\Gamma}$, and $[\bar{\boldsymbol{\eta}}(t)]_{1:N} = \bar{\mathbf{x}}(t)$, the update rule for $\mathbf{\Gamma}^{k+1} = \mathrm{diag}(\boldsymbol{\gamma}^{k+1})$ is derived by replacing $\mathbf{H}$, $\mathbf{\Phi}$ and $\bar{\eta}_n^k(t)$ in Eq. (55) with $\mathbf{\Gamma}$, $\mathbf{L}$ and $\bar{\mathbf{x}}_n^k(t)$, respectively, and defining the counterpart of $\mathbf{M}_{\mathrm{SN}}^k$ for sources accordingly as $\mathbf{M}_{\mathrm{S}}^k := \boldsymbol{\omega}_{\mathrm{S}}^k \mathbf{M}_{\mathrm{space}}^k (\boldsymbol{\omega}_{\mathrm{S}}^k)^\top$, where $\boldsymbol{\omega}_{\mathrm{S}}^k := \mathbf{\Gamma}^k \mathbf{L}^\top (\mathbf{\Sigma}_{\mathbf{y}}^k)^{-1}$. The update rule for the source variances is then obtained as follows:

$$
\gamma_n^{k+1} \leftarrow \sqrt{\frac{\left[\mathbf{M}_{\mathrm{S}}^k\right]_{n,n}}{\left[\mathbf{L}^\top \left(\mathbf{\Sigma}_{\mathbf{y}}^k\right)^{-1} \mathbf{L}\right]_{n,n}}} = \sqrt{\frac{\frac{1}{T}\sum_{t=1}^T (\bar{\mathbf{x}}_n^k(t))^2}{\mathbf{L}_n^\top \left(\mathbf{\Sigma}_{\mathbf{y}}^k\right)^{-1} \mathbf{L}_n}}
$$

$$
\text{for } n = 1, \ldots, N \,, \tag{56}
$$

where $\mathbf{L}_n$ denotes the $n$-th column of the lead field matrix.

### F.2 Update rule for noise variances

Similarly, given $[\mathbf{\Phi}]_{1:M,N+1:N+M} = \mathbf{I}$, $[\mathbf{H}]_{N+1:N+M,N+1:N+M} = \mathbf{\Lambda}$, and $[\bar{\boldsymbol{\eta}}(t)]_{N+1:N+M} = \bar{\mathbf{e}}(t) := \mathbf{y}(t) - \mathbf{L}\bar{\mathbf{x}}(t)$, the update rule for $\mathbf{\Lambda}^{k+1} = \mathrm{diag}(\boldsymbol{\lambda}^{k+1})$ is derived by replacing $\mathbf{H}$, $\mathbf{\Phi}$ and $\bar{\eta}_n^k(t)$ in Eq. (55) with $\mathbf{\Lambda}$, $\mathbf{I}$ and $\bar{\mathbf{e}}_n^k(t)$, respectively, and defining the counterpart of $\mathbf{M}_{\mathrm{SN}}^k$ for the noise accordingly as $\mathbf{M}_{\mathrm{N}}^k := \boldsymbol{\omega}_{\mathrm{N}}^k \mathbf{M}_{\mathrm{space}}^k (\boldsymbol{\omega}_{\mathrm{N}}^k)^\top$ with $\boldsymbol{\omega}_{\mathrm{N}}^k = \mathbf{\Lambda}^k (\mathbf{\Sigma}_{\mathbf{y}}^k)^{-1}$. The update rule for the noise variances is then derived as follows:

$$
\lambda_m^{k+1} \leftarrow \sqrt{\frac{\left[\mathbf{M}_{\mathrm{N}}^k\right]_{m,m}}{\left[\left(\mathbf{\Sigma}_{\mathbf{y}}^k\right)^{-1}\right]_{m,m}}} = \sqrt{\frac{\sum_{t=1}^T (\bar{\mathbf{e}}_n^k(t))^2}{\left[\left(\mathbf{\Sigma}_{\mathbf{y}}^k\right)^{-1}\right]_{m,m}}}
$$

$$
\text{for } m = 1, \ldots, M \,, \tag{57}
$$

which is identical to the update rule of the Champagne with heteroscedastic noise learning as presented in Cai et al. [68].

## G Proof of Theorem 5

We prove Theorem 5 by showing that the alternating update rules for $\mathbf{B}$ and $\mathbf{H}$, Eqs. (7) and (10), are guaranteed to converge to a stationary point of the Bayesian Type-II likelihood $\mathcal{L}_{\mathrm{kron}}(\mathbf{\Gamma}, \mathbf{\Lambda}, \mathbf{B})$ Eq. (4). More generally, we prove that full Dugh is an instance of the general class of majorization-minimization (MM) algorithms, for which this property follows by construction. To this end, we first briefly review theoretical concepts behind the majorization-minimization (MM) algorithmic framework [69–72].

### G.1 Required conditions for majorization-minimization algorithms

MM is a versatile framework for optimizing general non-linear optimization programs. The main idea behind MM is to replace the original cost function in each iteration by an upper bound, also known as majorizing function, whose minimum is easy to find. Compared to other popular optimization paradigms such as (quasi)-Newton methods, MM algorithms enjoy guaranteed convergence to a stationary point [27]. The MM class covers a broad range of common optimization algorithms such as *convex-concave procedures (CCCP)* and *proximal methods* [27, Section IV], [73–75]. Such algorithms have been applied in various domains such as non-negative matrix factorization [76], graph learning [77], robust portfolio optimization in finance [78], direction of arrival (DoA) and channel estimation in wireless communications [79–82], internet of things (IoT) [83, 84], and brain

source imaging [26, 68, 85–87]. Interested readers are referred to Sun et al. [27] for an extensive list of applications on MM.

We define an original optimization problem with the objective of minimizing a continuous function $f(\mathbf{u})$ within a closed convex set $\mathcal{U} \subset \mathbb{R}^n$:

$$\min_{\mathbf{u}} f(\mathbf{u}) \quad \text{subject to } \mathbf{u} \in \mathcal{U} . \tag{58}$$

Then, the idea of MM can be summarized as follows. First, construct a continuous *surrogate function* $g(\mathbf{u}|\mathbf{u}^k)$ that *majorizes*, or upper-bounds, the original function $f(\mathbf{u})$ and coincides with $f(\mathbf{u})$ at a given point $\mathbf{u}^k$:

$$[A1] \qquad g(\mathbf{u}^k|\mathbf{u}^k) = f(\mathbf{u}^k) \qquad \forall \, \mathbf{u}^k \in \mathcal{U}$$

$$[A2] \qquad g(\mathbf{u}|\mathbf{u}^k) \geq f(\mathbf{u}) \qquad \forall \, \mathbf{u}, \mathbf{u}^k \in \mathcal{U} .$$

Second, starting from an initial value $\mathbf{u}^0$, generate a sequence of feasible points $\mathbf{u}^1, \mathbf{u}^2, \ldots, \mathbf{u}^k, \mathbf{u}^{k+1}$ as solutions of a series of successive simple optimization problems, where

$$[A3] \qquad \mathbf{u}^{k+1} := \arg\min_{\mathbf{u} \in \mathcal{U}} g(\mathbf{u}|\mathbf{u}^k) .$$

**Definition 5.** *Any algorithm fulfilling conditions [A1]–[A3] is called a Majorization Minimization (MM) algorithm.*

If a surrogate function fulfills conditions [A1]–[A3], then the value of the cost function $f$ decreases in each iteration:

**Corollary 1.** *An MM algorithm has a* descending trend *property, whereby the value of the cost function $f$ decreases in each iteration: $f(\mathbf{u}^{k+1}) \leq f(\mathbf{u}^k)$.*

*Proof.* To verify the descending trend in the MM framework, it is sufficient to show that $f(\mathbf{u}^{k+1}) \leq f(\mathbf{u}^k)$. To this end, we have $f(\mathbf{u}^{k+1}) \leq g(\mathbf{u}^{k+1}|\mathbf{u}^k)$ from condition [A2]. Condition [A3] further states that $g(\mathbf{u}^{k+1}|\mathbf{u}^k) \leq g(\mathbf{u}^k|\mathbf{u}^k)$, while $g(\mathbf{u}^k|\mathbf{u}^k) = f(\mathbf{u}^k)$ holds according to [A1]. Putting everything together, we have:

$$f(\mathbf{u}^{k+1}) \overset{[A2]}{\leq} g(\mathbf{u}^{k+1}|\mathbf{u}^k) \overset{[A3]}{\leq} g(\mathbf{u}^k|\mathbf{u}^k) \overset{[A1]}{=} f(\mathbf{u}^k) ,$$

which concludes the proof. $\qquad\square$

While Corollary 1 guarantees a descending trend, convergence requires additional assumptions on particular properties of $f$ and $g$ [70, 71]. For the smooth functions considered in this paper, we require that the derivatives of the original and surrogate functions coincide at $\mathbf{u}^k$:

$$[A4] \qquad \nabla g(\mathbf{u}^k|\mathbf{u}^k) = \nabla f(\mathbf{u}^k) \qquad \forall \, \mathbf{u}^k \in \mathcal{U} .$$

We can then formulate the following, stronger, theorem:

**Theorem 8.** *For an MM algorithm that additionally satisfies [A4], every limit point of the sequence of minimizers generated through [A3] is a stationary point of the original optimization problem Eq. (58).*

*Proof.* A detailed proof is provided in Razaviyayn et al. [70, Theorem 1]. $\qquad\square$

Note that since we are working with smooth functions, conditions [A1]–[A4] are sufficient to prove convergence to a stationary point according to Theorem 8.

### G.2 Detailed derivation of the proof of Theorem 5

We now show that full Dugh is an instance of majorization-minimization as defined above, which fulfills Theorem 8.

*Proof.* We need to prove that conditions [A1]–[A4] are fulfilled for full Dugh. To this end, we first prove conditions [A1]–[A4] for the optimization with respect to $\mathbf{B}$ based on the convex surrogate function in Eq. (5), $\mathcal{L}_{\text{conv}}^{\text{time}}(\boldsymbol{\Gamma}^k, \boldsymbol{\Lambda}^k, \mathbf{B})$. For this purpose, we recall the upper bound on $\log |\mathbf{B}|$ in Eq. (28), which fulfills condition [A2] since it majorizes $\log |\mathbf{B}|$ as a result of the concavity of the log-determinant function and its first-order Taylor expansion around $\mathbf{B}^k$. Besides, it automatically satisfies conditions [A1] and [A4] by construction, because the majorizing function in Eq. (28) is obtained through a Taylor expansion around $\mathbf{B}^k$. Concretely, [A1] is satisfied because the equality in Eq. (28) holds for $\mathbf{B} = \mathbf{B}^k$. Similarly, [A4] is satisfied because the gradient of $\log |\mathbf{B}|$ at point $\mathbf{B}^k$, $\left(\mathbf{B}^k\right)^{-1}$ defines the linear Taylor approximation $\log |\mathbf{B}^k| + \text{tr}\left(\left(\mathbf{B}^k\right)^{-1}\left(\mathbf{B} - \mathbf{B}^k\right)\right)$. Thus, both gradients coincide in $\mathbf{B}^k$ by construction. We can further prove that [A3] can be satisfied by showing that $\mathcal{L}_{\text{conv}}^{\text{time}}(\boldsymbol{\Gamma}^k, \boldsymbol{\Lambda}^k, \mathbf{B})$ reaches its global minimum in each MM iteration. This is guaranteed if $\mathcal{L}_{\text{conv}}^{\text{time}}(\boldsymbol{\Gamma}^k, \boldsymbol{\Lambda}^k, \mathbf{B})$ can be shown to be convex or g-convex with respect to $\mathbf{B}$. To this end, we first require the subsequent proposition:

**Proposition 2.** *Any local minimum of a g-convex function over a g-convex set is a global minimum.*

*Proof.* A detailed proof is presented in Rapcsak [57, Theorem 2.1]. $\square$

Given the proof presented in Appendix C.1, we can conclude that $\mathcal{L}_{\text{conv}}^{\text{time}}(\mathbf{H}^k, \mathbf{B})$ is g-convex; hence, any local minimum of $\mathcal{L}_{\text{conv}}^{\text{time}}(\mathbf{H}^k, \mathbf{B})$ is a global minimum according to Proposition 2. This proves that condition [A3] is fulfilled and completes the proof that the optimization of Eq. (4) with respect to $\mathbf{B}$ using the convex surrogate cost function Eq. (5) leads to an MM algorithm.

The proof of conditions [A1], [A2] and [A4] for the optimization with respect to $\mathbf{H}$ based on the convex surrogate function in Eq. (8), $\mathcal{L}_{\text{conv}}^{\text{space}}(\mathbf{H}, \mathbf{B}^k)$, can be presented analogously. To this end, we recall the upper bound on $\log |\boldsymbol{\Sigma_y}|$ in Eq. (41), which fulfills condition [A2] since it majorizes $\log |\boldsymbol{\Sigma_y}|$ as a result of the concavity of the log-determinant function and its first-order Taylor expansion around $\boldsymbol{\Sigma_y^k}$. Besides, it automatically satisfies conditions [A1] and [A4] by construction, because the majorizing function in Eq. (41) is obtained through a Taylor expansion around $\boldsymbol{\Sigma_y^k}$. Concretely, [A1] is satisfied because the equality in Eq. (41) holds for $\boldsymbol{\Sigma_y} = \boldsymbol{\Sigma_y^k}$. Similarly, [A4] is satisfied because the gradient of $\log |\boldsymbol{\Sigma_y}|$ at point $\boldsymbol{\Sigma_y^k}$, $\left(\boldsymbol{\Sigma_y^k}\right)^{-1}$ defines the linear Taylor approximation $\log |\boldsymbol{\Sigma_y^k}| + \text{tr}\left[\left(\boldsymbol{\Sigma_y^k}\right)^{-1}\left(\boldsymbol{\Sigma_y} - \boldsymbol{\Sigma_y^k}\right)\right]$. Thus, both gradients coincide in $\boldsymbol{\Sigma_y^k}$ by construction. We can further prove that [A3] can be satisfied by showing that $\mathcal{L}_{\text{conv}}^{\text{time}}(\mathbf{H}^k, \mathbf{B})$ reaches its global minimum in each MM iteration. This is guaranteed if $\mathcal{L}_{\text{conv}}^{\text{time}}(\mathbf{H}^k, \mathbf{B})$ can be shown to be convex with respect to $\mathbf{H} = \text{diag}(\mathbf{h})$. Given the proof presented in Appendix E.2, we can show that [A3] is also satisfied since $\mathcal{L}_{\text{conv}}^{\text{space}}(\mathbf{H}, \mathbf{B}^k)$ in Eq. (8) is a convex function with respect to $\mathbf{h}$. The convexity of $\mathcal{L}_{\text{conv}}^{\text{space}}(\mathbf{H}, \mathbf{B}^k)$, which ensures that condition [A3] can be satisfied using standard optimization, along with the fulfillment of conditions [A1], [A2] and [A4], ensure that Theorem 8 holds for $\mathcal{L}_{\text{conv}}^{\text{space}}(\mathbf{H}, \mathbf{B}^k)$. This completes the proof that the optimization of Eq. (4) with respect to $\mathbf{H}$ using the convex surrogate cost function Eq. (8) leads to an MM algorithm that is guaranteed to converge. $\square$

# H    Proof of Proposition 1

This is a well-established classical results of the signal processing literature. Therefore, we only provide two remarks highlighting important connections between Preposition 1 and our proposed method, and refer the interested reader to Grenander and Szegö [32, Chapter 5] for a detailed proof of the theorem.

**Remark 3.** *As indicated in Babu [88], the set of embedded circulant matrices with size $L \times L$, $\mathcal{B}^L$, does not constist exclusively of positive definite Toeplitz matrices. Therefore, we restrict ourselves to embedded circulant matrices $\mathbf{P}$ that are also positive definite. This restriction indeed makes the diagonalization inaccurate, but we can improve the accuracy by choosing a large $L$. Practical evaluations have shown that choosing $L \geq 2T - 1$ provides sufficient approximation result.*

**Remark 4.** *The Carathéodory parametrization of a Toeplitz matrix [89, Section 4.9.2] states that any PD matrix can be represented as $\mathbf{B} = \mathbf{A}\mathbf{P}'\mathbf{A}^H$ with $[\mathbf{A}]_{m,l} = e^{i(w_l)(m-1)}$ and $\mathbf{P}' = \text{diag}(p_0', p_1', \dots, p_{L-1}')$ where $w_l$ and $p_l$ are specific frequencies and their corresponding*

*amplitudes. By comparing the* Carathéodory parametrization *of* $\mathbf{B}$ *with its Fourier diagonalization (Eq.* (15)*), it can be seen that Fourier diagonalization force the frequencies to lie on the Fourier grid, i.e.* $w_l = \frac{2\pi(l-1)}{L}$ *, which indeed makes the diagonalization slightly inaccurate. The approximation accuracy can, however, be improved by increasing L. The Szegö theorem* [32, 90] *states that a Toeplitz matrix is asymptotically* $(L \to \infty)$ *diagonalized by the DFT matrix.*

# I  Proof of Theorem 6

*Proof.* We proceed in two steps. First, we show that the cost function in Eq. (13) is convex with respect to $\mathbf{p}$. In the second step, we then derive the update rule proposed in (16).

## I.1  Part I: Convexity of the majorizing cost function

The proof of this section parallels the one provided in [91, Proposition 4]. We start by recalling Eq. (13):

$$\mathbf{B}^* = \underset{\mathbf{B} \in \mathcal{B},\ \mathbf{H} = \mathbf{H}^k}{\arg\min}\ \operatorname{tr}((\mathbf{B}^k)^{-1}\mathbf{B}) + \operatorname{tr}(\mathbf{M}_{\text{time}}^k \mathbf{B}^{-1})\ . \tag{59}$$

We then show that the second term in Eq. (59) can be upper-bounded as follows:

$$\operatorname{tr}(\mathbf{M}_{\text{time}}^k \mathbf{B}^{-1}) \le \operatorname{tr}(\mathbf{P}^k \mathbf{Q}^H (\mathbf{B}^k)^{-1} \mathbf{M}_{\text{time}}^k (\mathbf{B}^k)^{-1} \mathbf{Q} \mathbf{P}^k \mathbf{P}^{-1})\ . \tag{60}$$

By defining $\mathbf{V}$ as

$$\mathbf{V} = \begin{bmatrix} (\mathbf{B}^k)^{-1} \mathbf{Q} \mathbf{P}^k \mathbf{P}^{\frac{-1}{2}} \\ \mathbf{Q} \mathbf{P}^{\frac{1}{2}} \end{bmatrix}\ , \tag{61}$$

the PSD property of $\mathbf{S}$ can be inferred as

$$\mathbf{S} = \begin{bmatrix} (\mathbf{B}^k)^{-1} \mathbf{Q} \mathbf{P}^k \mathbf{P}^{-1} \mathbf{P}^k \mathbf{Q}^H (\mathbf{B}^k)^{-1} & \mathbf{I} \\ \mathbf{I} & \mathbf{Q} \mathbf{P} \mathbf{Q}^H \end{bmatrix} = \mathbf{V} \mathbf{V}^H \ge 0\ . \tag{62}$$

Therefore by virtue of the Schur complement with $\mathbf{D} = (\mathbf{B}^k)^{-1} \mathbf{Q} \mathbf{P}^k \mathbf{P}^{-1} \mathbf{P}^k \mathbf{Q}^H (\mathbf{B}^k)^{-1}$, $\mathbf{G} = \mathbf{I}$ and $\mathbf{B} = \mathbf{Q} \mathbf{P} \mathbf{Q}^H$, we have:

$$(\mathbf{B}^k)^{-1} \mathbf{Q} \mathbf{P}^k \mathbf{P}^{-1} \mathbf{P}^k \mathbf{Q}^H (\mathbf{B}^k)^{-1} \ge \left( \mathbf{Q} \mathbf{P} \mathbf{Q}^H \right)^{-1}\ . \tag{63}$$

The inequality (Eq. (60)) can be directly obtained by multiplying $\mathbf{M}_{\text{time}}^k$ to both sides of Eq. (63), applying the trace operator, using Eq. (14) and finally rearranging the terms within the trace operator:

$$\operatorname{tr}(\mathbf{M}_{\text{time}}^k (\mathbf{B}^k)^{-1} \mathbf{Q} \mathbf{P}^k \mathbf{P}^{-1} \mathbf{P}^k \mathbf{Q}^H (\mathbf{B}^k)^{-1}) \ge \operatorname{tr}(\mathbf{M}_{\text{time}}^k \mathbf{B}^{-1})\ . \tag{64}$$

Let $\mathbf{B}_k = \mathbf{Q} \mathbf{P}^k \mathbf{Q}^H$ be the Fourier diagonalization of a fixed matrix $\mathbf{B}_k$ in the $k$-th iteration, one can derive an efficient update rule for the temporal covariance by rewriting Eq. (13) and exploiting Propositions 1 and Eq. (60):

$$\operatorname{tr}((\mathbf{B}^k)^{-1}\mathbf{B}) + \operatorname{tr}(\mathbf{B}^{-1}\mathbf{M}_{\text{time}}^k)$$
$$\le \operatorname{tr}((\mathbf{B}^k)^{-1}\mathbf{Q}\mathbf{P}\mathbf{Q}^H) + \operatorname{tr}(\mathbf{P}^k \mathbf{Q}^H (\mathbf{B}^k)^{-1} \mathbf{M}_{\text{time}}^k (\mathbf{B}^k)^{-1} \mathbf{Q} \mathbf{P}^k \mathbf{P}^{-1})$$
$$= \operatorname{diag}(\mathbf{Q}^H (\mathbf{B}^k)^{-1} \mathbf{Q})\mathbf{p} + \operatorname{diag}(\mathbf{P}^k \mathbf{Q}^H (\mathbf{B}^k)^{-1} \mathbf{M}_{\text{time}}^k (\mathbf{B}^k)^{-1} \mathbf{Q} \mathbf{P}^k)\mathbf{p}^{-1}\ , \tag{65}$$

where $\mathbf{p} = \operatorname{vec}(\mathbf{P})$, and $\mathbf{p}^{-1}$ is defined as the element-wise inversion of $\mathbf{p}$.

We formulate the optimization problem as follows:

$$\mathcal{L}_{\text{toeplitz}}^{\text{time}}(\mathbf{p}) = \operatorname{diag}(\mathbf{Q}^H (\mathbf{B}^k)^{-1} \mathbf{Q})\mathbf{p}$$
$$+ \operatorname{diag}(\mathbf{P}^k \mathbf{Q}^H (\mathbf{B}^k)^{-1} \mathbf{M}_{\text{time}}^k (\mathbf{B}^k)^{-1} \mathbf{Q} \mathbf{P}^k)\mathbf{p}^{-1}\ . \tag{66}$$

Let $\mathbf{W}^k := \mathbf{Q}^H (\mathbf{B}^k)^{-1} \mathbf{Q}$ and $\mathbf{O}^k := \mathbf{P}^k \mathbf{Q}^H (\mathbf{B}^k)^{-1} \mathbf{M}_{\text{time}}^k (\mathbf{B}^k)^{-1} \mathbf{Q} \mathbf{P}^k$. Then, we rewrite $\mathcal{L}_{\text{toeplitz}}^{\text{time}}(\mathbf{p})$ as

$$\mathcal{L}_{\text{toeplitz}}^{\text{time}}(\mathbf{p}) = \operatorname{diag}(\mathbf{W}^k)\mathbf{p} + \operatorname{diag}(\mathbf{O}^k)\mathbf{p}^{-1}\ . \tag{67}$$

The convexity of $\mathcal{L}_{\text{toeplitz}}^{\text{time}}(\mathbf{p})$ can be directly inferred from the convexity of $\operatorname{diag}\left[\mathbf{W}^k\right]\mathbf{p}$ and $\operatorname{diag}\left[\mathbf{O}^k\right]\mathbf{p}^{-1}$ with respect to $\mathbf{p}$ [67, Chapter. 3].

## I.2   Part II: Derivation of the update rule in Eq. (16)

We now present the second part of the proof by deriving the update rule in Eq. (16). Since the cost function $\mathcal{L}_{\text{toeplitz}}^{\text{time}}(\mathbf{p})$ is convex, its optimal solution in the $k$-th iteration is unique. More concretely, a closed-form solution of the final update rule can be obtained by taking the derivative of Eq. (67) with respect to $\mathbf{p}$ and setting it to zero:

$$p_l^{k+1} \leftarrow \sqrt{\frac{\hat{g}_l^k}{\hat{z}_l^k}} \text{ for } l = 0, \ldots, L-1 \text{, where} \tag{68}$$

$$\hat{\mathbf{g}} = \operatorname{diag}(\mathbf{P}^k \mathbf{Q}^H (\mathbf{B}^k)^{-1} \mathbf{M}_{\text{time}}^k (\mathbf{B}^k)^{-1} \mathbf{Q} \mathbf{P}^k) \tag{69}$$

$$\hat{\mathbf{z}} = \operatorname{diag}(\mathbf{Q}^H (\mathbf{B}^k)^{-1} \mathbf{Q}) \,, \tag{70}$$

which concludes the proof. $\qquad\square$

## J   Proof of Theorem 7

*Proof.* The proof is inspired by ideas presented in Rakitsch et al. [2], Wu et al. [49], Saatçi [92] for spatio-temporal Gaussian process inference, and parallels the one proposed in Solin et al. [93]. Rakitsch et al. [2], Wu et al. [49] provide an efficient method for computing the non-convex spatio-temporal ML cost function by exploiting the compatibility between diagonalization and the Kronecker product. Here we use similar ideas to obtain the posterior mean in an efficient way.

Recalling the diagonalization of the temporal correlation matrix as $\mathbf{B} = \mathbf{Q}\mathbf{P}\mathbf{Q}^H$ and considering the eigenvalue decomposition of $\mathbf{L}\boldsymbol{\Gamma}\mathbf{L}^\top$ as $\mathbf{L}\boldsymbol{\Gamma}\mathbf{L}^\top = \mathbf{U_x}\mathbf{D_x}\mathbf{U_x}^\top$ with $\mathbf{D_x} = \operatorname{diag}(d_1, \ldots, d_M)$, we have:

$$
\begin{aligned}
\bar{\mathbf{x}}_g &= (\boldsymbol{\Gamma} \otimes \mathbf{B}) \, \mathbf{D}^\top \tilde{\boldsymbol{\Sigma}}_{\mathbf{y}}^{-1} \mathbf{y}_g \\
&= (\boldsymbol{\Gamma} \otimes \mathbf{B}) \, (\mathbf{L} \otimes \mathbf{I})^\top \left( \boldsymbol{\Lambda} \otimes \mathbf{B} + \mathbf{D}\boldsymbol{\Sigma}_0 \mathbf{D}^\top \right)^{-1} \operatorname{vec}(\mathbf{Y}_g^\top) \\
&= (\boldsymbol{\Gamma} \otimes \mathbf{B}) \, (\mathbf{L}^\top \otimes \mathbf{I}) \left( (\boldsymbol{\Lambda} + \mathbf{L}\boldsymbol{\Gamma}\mathbf{L}^\top) \otimes \mathbf{B} \right)^{-1} \operatorname{vec}(\mathbf{Y}_g^\top) \\
&= \left( \boldsymbol{\Gamma}\mathbf{L}^\top \otimes \mathbf{B} \right) \left( (\boldsymbol{\Lambda} + \mathbf{L}\boldsymbol{\Gamma}\mathbf{L}^\top) \otimes \mathbf{B} \right)^{-1} \operatorname{vec}(\mathbf{Y}_g^\top) \\
&= \left( \boldsymbol{\Gamma}\mathbf{L}^\top \mathbf{U_x} \otimes \mathbf{Q}\mathbf{P} \right) (\boldsymbol{\Omega})^{-1} (\mathbf{U_x}^\top \otimes \mathbf{Q}^H) \operatorname{vec}(\mathbf{Y}_g^\top) \\
&= \left( \boldsymbol{\Gamma}\mathbf{L}^\top \mathbf{U_x} \otimes \mathbf{Q}\mathbf{P} \right) (\boldsymbol{\Omega})^{-1} \operatorname{tr} \left( \mathbf{Q}^H \mathbf{Y}_g^\top \mathbf{U_x} \right) \\
&= \left( \boldsymbol{\Gamma}\mathbf{L}^\top \mathbf{U_x} \otimes \mathbf{Q}\mathbf{P} \right) \operatorname{tr} \left( \boldsymbol{\Pi} \odot \mathbf{Q}^H \mathbf{Y}_g^\top \mathbf{U_x} \right) \\
&= \operatorname{tr} \left( \mathbf{Q}\mathbf{P} \left( \boldsymbol{\Pi} \odot \mathbf{Q}^H \mathbf{Y}_g^\top \mathbf{U_x} \right) \left( \mathbf{U_x}^\top \mathbf{L}\boldsymbol{\Gamma}^\top \right) \right) \,,
\end{aligned}
\tag{71}
$$

where $\odot$ denotes the Hadamard product between corresponding elements of two matrices. $\boldsymbol{\Omega}$ and $\boldsymbol{\Pi}$ are defined as follows: $\boldsymbol{\Omega} = \boldsymbol{\Lambda} + \mathbf{D_x} \otimes \mathbf{P}$ and $[\boldsymbol{\Pi}]_{l,m} = \frac{1}{\sigma_m^2 + p_l d_m}$ for $l = 1, \ldots, L$; $m = 1, \ldots, M$. Note that the last four lines are derived based on the following matrix equality:

$$\operatorname{tr}(\mathbf{A}^\top \mathbf{B}\mathbf{C}\mathbf{D}^\top) = \operatorname{vec}(\mathbf{A})^\top (\mathbf{D} \otimes \mathbf{B}) \operatorname{vec}(\mathbf{C}). \tag{72}$$

Together with the update rule in Eq. (19), this concludes the proof of Theorem 7. $\qquad\square$

## K   Details on the simulation set-up

### K.1   Forward modeling

Populations of pyramidal neurons in the cortical gray matter are known to be the main drivers of the EEG signal [14, 19]. Here, we use a realistic volume conductor model of the human head to model the linear relationship between primary electrical source currents generated within these populations and the resulting scalp surface potentials captured by EEG electrodes. The lead field matrix, $\mathbf{L} \in \mathbb{R}^{58 \times 2004}$, which serves as the forward model in our simulations, was generated using the New York Head model [36]. The New York Head model provides a segmentation of an average human head into six different tissue types, taking into account the realistic anatomy and electrical tissue conductivities. In this model, 2004 dipolar current sources were placed evenly on the cortical

surface and 58 sensors were placed on the scalp according to the extended 10-20 system [94]. Finally, the lead field matrix was computed using the finite element method (FEM) for a given head geometry and exploiting the quasi-static approximation of Maxwell's equations [14, 19, 36, 95].

Note that in accordance with the predominant orientation of pyramidal neuron assemblies, the orientation of all simulated source currents was fixed to be perpendicular to the cortical surface, so that only the scalar deflection of each source along the fixed orientation needs to be estimated. In real data analyses in Section 6 and Appendix L, however, surface normals are hard to estimate or even undefined in case of volumetric reconstructions. Consequently, we model each source in real data analyses as a full 3-dimensional current vector. This is achieved by introducing three variance parameters for each source within the source covariance matrix, $\mathbf{\Gamma}^{\text{3D}} = \text{diag}(\boldsymbol{\gamma}^{\text{3D}}) = [\gamma_1^x, \gamma_1^y, \gamma_1^z, \ldots, \gamma_N^x, \gamma_N^y, \gamma_N^z]^\top$. As all algorithms considered here model the source covariance matrix $\mathbf{\Gamma}$ to be diagonal, this extension can be readily implemented. Correspondingly, a full 3D leadfield matrix, $\mathbf{L}^{\text{3D}} \in \mathbb{R}^{M \times 3N}$, is used.

### K.2   Pseudo-EEG signal generation

We simulated a sparse set of $N_0 = 3$ active sources, which were placed at random positions on the cortex. To simulate the electrical neural activity of these sources, $T = \{10, 20, 50, 100\}$ time points were sampled from a univariate linear autoregressive (AR) process, which models the activity at time $t$ as a linear combination of the $P$ past values:

$$x_i(t) = \sum_{p=1}^{P} a_i(p) x_i(t-p) + \xi_i(t), \text{ for } i = 1, 2, 3 . \tag{73}$$

Here, $a_i(p)$ for $i = 1, 2, 3$ are linear AR coefficients, and $P$ is the order of the AR model. The model residuals $\xi_i(\cdot)$ for $i = 1, 2, 3$ are also referred to as the innovation process; their variance determines the stability of the overall AR process. We here assume uncorrelated standard normal distributed innovations, which are independent for all sources. In this experiment, we use stable AR systems of order $P = \{1, 2, 5, 7\}$. The resulting source distribution, represented as $\mathbf{X} = [\mathbf{x}(1), \ldots, \mathbf{x}(T)]$, was projected to the EEG sensors through application of lead field matrix: $\mathbf{Y}^{\text{signal}} = \mathbf{LX}$. Next we added Gaussian white noise to the sensor space signal. To this end, the same number of data points as the sources were sampled from a zero-mean normal distribution, where the time points assumed to be independent and identically distributed. The resulting noise distribution, represented as $\mathbf{E} = [\mathbf{e}(1), \ldots, \mathbf{e}(T)]$, is then normalized by its Frobenius norm and added to the signal matrix $\mathbf{Y}^{\text{signal}}$ as follows: $\mathbf{Y} = \mathbf{Y}^{\text{signal}} + \frac{(1-\alpha)\|\mathbf{Y}^{\text{signal}}\|_F}{\alpha\|\mathbf{E}\|_F}\mathbf{E}$, where $\alpha$ determines the signal-to-noise ratio (SNR) in sensor space. Precisely, SNR is defined as follows: $\text{SNR} = 20\log_{10}(\alpha/1-\alpha)$. In this experiment the following values of $\alpha$ were used: $\alpha = \{0.55, 0.65, 0.7, 0.8\}$, which correspond to the following SNRs: SNR$=\{1.7, 5.4, 7.4, 12\}$ (dB). Interested reader can refer to Haufe and Ewald [37] for a more details on this simulation framework.

### K.3   Source reconstruction and evaluation metrics

We applied thin Dugh to the synthetic datasets described above. In addition to thin Dugh, one further Type-II Bayesian learning scheme, namely Champagne [33], and two Type-I source reconstruction schemes, namely S-FLEX [35] and eLORETA [34], were also included as benchmarks with respect to source reconstruction performance. S-FLEX is used as an example of a sparse Type-I Bayesian learning method based on $\ell_1$-norm minimization. As spatial basis functions, unit impulses were used, so that the resulting estimate was identical to the so-called minimum-current estimate (MCE) [96]. eLORETA estimate, as an example of a smooth inverse solution based on weighted $\ell_2^2$-norm minimization, was used with $5\%$ regularization, whereas S-FLEX was fitted so that the residual variance was consistent with the ground-truth noise level. Note that the $5\%$ rule is chosen as it gives the best performance across a subset of regularization values ranging between $0.5\%$ to $15\%$. For thin Dugh, the noise variances as well as the variances of all voxels were initialized randomly by sampling from a standard normal distribution. The optimization program was terminated after reaching convergence. Convergence was defined if the relative change of the Frobenius-norm of the reconstructed sources between subsequent iterations was less than $10^{-8}$. A maximum of 1000 iterations was carried out if no convergence was reached beforehand.

Source reconstruction performance was evaluated according to two different measures, the *earth mover's distance* (EMD), used to quantify the spatial localization accuracy, and the correlation

between the original and reconstructed sources, $\hat{\mathbf{X}}$ and $\mathbf{X}$. The EMD metric measures the cost needed to map two probability distributions, defined on the same metric domain, into each other, see [21, 38]. It was applied here to the power of the true and estimated source activations defined on the cortical surface of the brain, which were obtained by taking the voxel-wise $\ell_2$-norm along the time domain. EMD was normalized to $[0, 1]$. The correlation between simulated and reconstructed source time courses was assessed as the mean of the absolute correlations obtained for each source, after optimally matching simulated and reconstructed sources. To this end, Pearson correlation between all pairs of simulated and reconstructed (i.e., those with non-zero activations) sources was measured. Each simulated source was matched to a reconstructed source based on maximum absolute correlation. Time-course correlation error (TCE) was then defined as one minus the average of these absolute correlations across sources. Each simulation was carried out 100 times using different instances of $\mathbf{X}$ and $\mathbf{E}$, and the mean and standard error of the mean (SEM) of each performance measure across repetitions was calculated.

# L   Real data analysis

## L.1   Auditory and visual evoked fields (AEF and VEF)

The MEG data used in this article were acquired in the Biomagnetic Imaging Laboratory at the University of California San Francisco (UCSF) with a CTF Omega 2000 whole-head MEG system from VSM MedTech (Coquitlam, BC, Canada) with 1200 Hz sampling rate. The neural responses for one subject's auditory evoked fields (AEF) and visual evoked fields (VEF) were localized. The AEF response was elicited while subjects were passively listening to 600 ms duration tones (1 kHz) presented binaurally. Data from 120 trial epochs were analysed. The VEF response was elicited while subjects were viewing images of objects projected onto a screen and subjects were instructed to name the objects verbally. Both AEF and VEF data were first digitally filtered from 1 to 70 Hz to remove artifacts and DC offset, time-aligned to the stimulus. Different number of trials were included for algorithm analyses. The pre-stimulus window was selected to be $-100$ ms to $-5$ ms and the post-stimulus time window was selected to be 60 ms to 180 ms, where 0 ms is the onset of the tone. The lead field for each subject was calculated with NUTMEG [97] using a single-sphere head model (two spherical orientation lead fields) and an 8 mm voxel grid. Each column was normalized to have a norm of unity. Further details on these datasets can be found in [33, 98–100].

Figure 6 shows the reconstructed sources of the Auditory Evoked Fields (AEF) from a representative subject using Champagne, thin and full Dugh. In this case, we tested the reconstruction performance of all algorithms with the number of trials limited to 20 and 120. As Figure 6 demonstrates, the performance of Dugh remains robust as the number of trials is increased to 20 and 120 in Figure 6. Finally, the VEF performance of benchmark algorithm eLORETA is demonstrated in Figure 7.

## L.2   EEG Data: Faces vs scrambled pictures

A publicly available EEG dataset (128-channel Biosemi ActiveTwo system) was downloaded from the SPM website (http://www.fil.ion.ucl.ac.uk/spm/data/mmfaces) and the lead field was calculated in SPM8 using a three-shell spherical model at the coarse resolution of 5124 voxels at approximately 8 mm spacing. These EEG data were also obtained during a visual response paradigm that involved randomized presentation of at least 86 faces and 86 scrambled faces. To examine the differential responses to faces across all trials, the averaged responses to scrambled-faces were subtracted from the averaged responses to faces. The result is demonstrated in Figure 8.

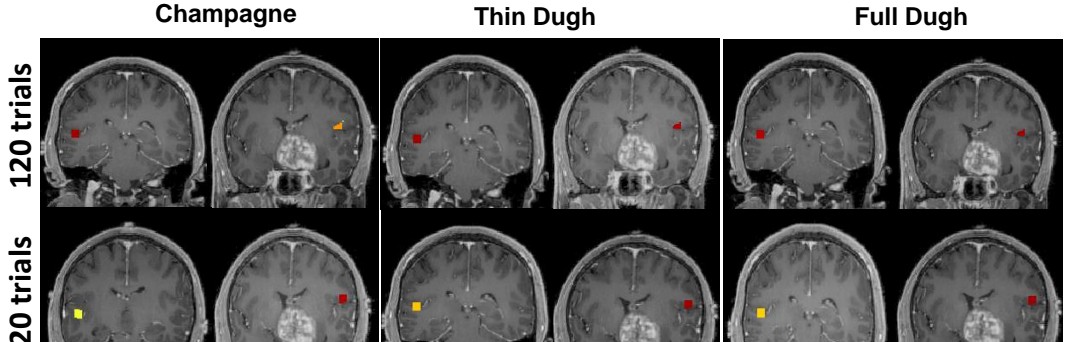

Figure 6: Robustness of Dugh and Champagne performance when the number of trials is increased to 20 and 120.

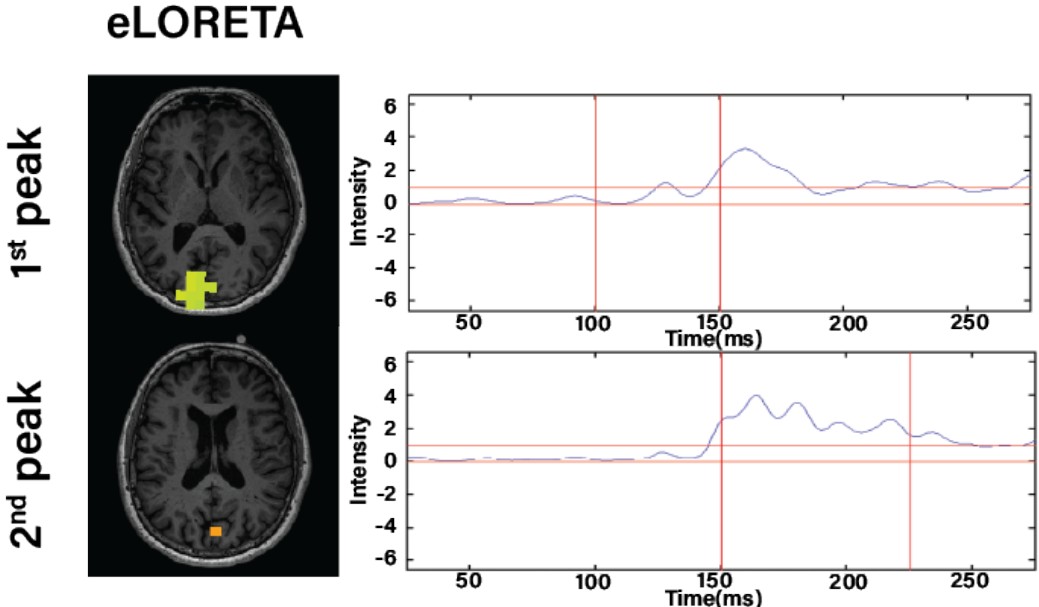

Figure 7: VEF performance of benchmark algorithm eLORETA. This benchmark did not yield reliable results for 5 trial epochs. Even when the number of trials were increased to 20, benchmark's performance yielded neither good spatial localization of the two visual cortical areas nor good estimation of the time courses of these activations.

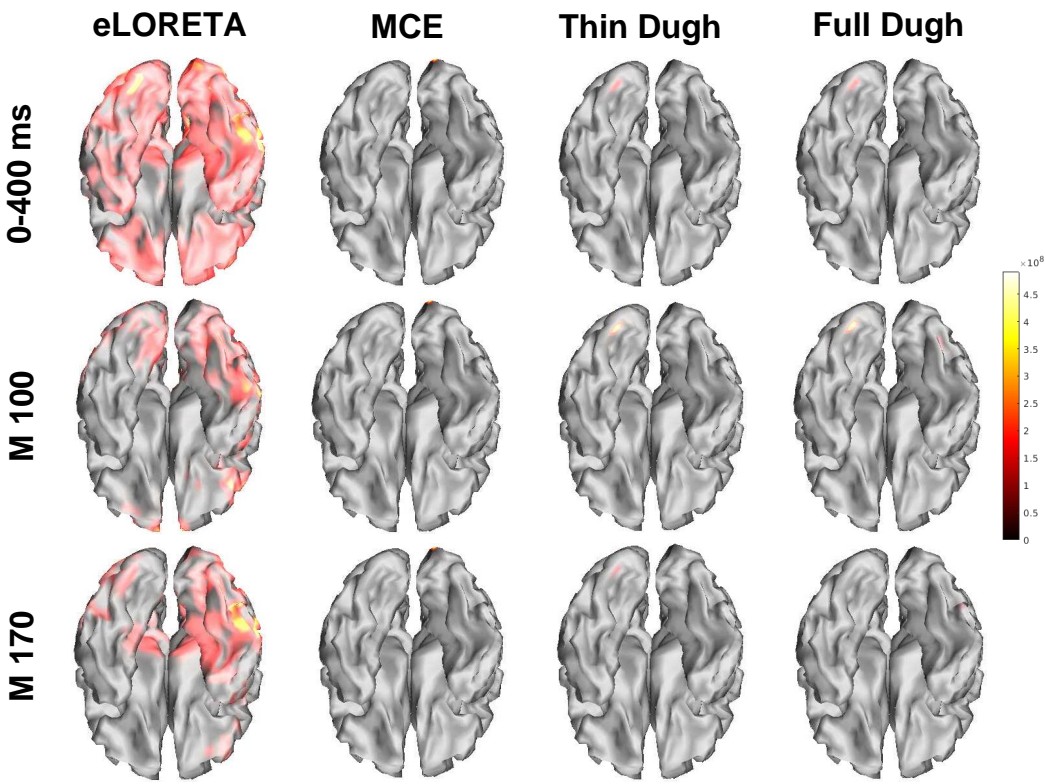

Figure 8: Performance of Dugh and benchmarks on EEG data acquired during a face recognition task. Dugh was able to provide more focal and distinct activations for the M100 and M170 responses that were not clearly identified using the benchmarks eLORETA and MCE.