# OpenReview forum: "Efficient hierarchical Bayesian inference for spatio-temporal regression models in neuroimaging"
_NeurIPS.cc/2021/Conference — NeurIPS 2021 Poster_

### Official Review · Reviewer_KaNL · 2021-06-30

**Rating:** 7
**Confidence:** 3

**Summary:**

The paper proposes a Bayesian framework for source estimation in M/EEG. The Bayesian framework explicitly estimates spatial and temporal covariance matrices for both signal and noise, before estimating the posterior mean. A majorization-minimization algorithm is proposed to estimate the model, and detailed proofs are provided, including a convergence proof. The model is benchmarked against three source-estimation methods on simulated signals. Finally, the model is used in empirical recordings with a challenging low number of trials.

**Limitations And Societal Impact:**

Some discussion is provided about the limitations of the model assumptions.

Reporting computational cost of each method in the experiments would be appreciated.

Societal impact is not discussed, although it is probably not necessary.

**Main Review:**

**Originality**: The problem of M/EEG source estimation is a well-known and challenging problem. This paper propose a new Bayesian model to solve this problem. The novelty resides in the explicit spatial and temporal priors, where the covariances are estimated jointly with the model. Related work seems to be well cited and discussed.

**Quality**: The model is sound and technically advanced, and the methods used are appropriate. The technical derivations (in supplementary) are thorough and well described. The proofs look reasonable, though I am not familiar with this sort of work, and I have only skimmed through them. The simulation benchmark is short, and only consider a subset of the competing methods, but results are convincing. The empirical result seems impressive. The discussion of the pros and cons of this method is appreciated, although the discussion against type-I methods could have been supported by a benchmark.

**Clarity**: The submission is clearly written and well organized. A large part of the work is located in the long supplementary document, with detailed proofs and discussions. The experimental part is well described, although less well polished than the technical derivations. The figure quality is mediocre. Importantly, the empirical results (Figure 3) are not interpretable without the help of Figure 4-5-6 (all in supplementary). Improving the organization of these figures could help the reader understand the interest of these results.

**Significance**: The model is sounds and seems to be more robust than other models. I am not able to judge if it is a game changer, but it does advance the state of the art in this domain, although the M/EEG practitioners often ignore new advanced models and only use simple ones. To make this work actually used by the community, an effort should be made to release the code with extensive documentation and examples, ideally in Python (the shared code is in a proprietary language (matlab)).

minor comments:
- hyperlinks to references do not work
- L44: M, N and T are not defined
- Fig1: really low resolution

**Time Spent Reviewing:**

3

---

> ### Author Response · Authors · 2021-08-10
> **Response to Reviewer KaNL**
>
> ## Response to “Summary” and  “Main Review”:
>
> We thank the reviewer for a concise summary of the paper and constructive and useful comments as well as helpful feedback.
>
> ## Response to “Clarity”:
> We would be happy to improve the figure quality and integrate figures 3-6 into the main body in a revision while adhering to the space constraints. We would also be glad to implement any specific suggestion from the reviewer during this discussion period.
>
> ## Response to “Significance”:
> We are already sharing our MATLAB codes. In parallel, we are working with other developers to make these codes available in Python, specifically within the MNE python package suitable for EEG/MEG analysis. This Python package will certainly be accompanied with informative and comprehensive documentations.
>
> ### Response to “minor comments”:
> Thank you for pointing out these typos. We will fix those in a revision.
>
> ## Response to “Limitations And Societal Impact”:
> Regarding the computational cost of each method, we would certainly be glad to include some information regarding runtime comparisons of accuracy and efficiency of the convergence of loss-functions for each method either in the main body of the paper or in the supplementary materials.

---

> > ### Comment · Reviewer_KaNL · 2021-08-24
> > **Thanks**
> >
> > Adding the computational cost of each tested method, along with a Matlab and Python code will be very valuable.
> > The authors adequately answered my concerns.

---

### Official Review · Reviewer_bAFS · 2021-07-14

**Rating:** 6
**Confidence:** 3

**Summary:**

This manuscript presents an efficient majorization-minimization optimization framework to infer the parameters of the spatio-temporal covariance matrix in the MEEG signal source reconstruction problem. The main contribution lies in simplifying a non-convex optimization problem using a convex surrogate function with a convergence guarantee. The method has been tested in few limited scenarios on simulated and real MEEG data and the results are compared against three alternative approaches.


**Limitations And Societal Impact:**

The proposed method in this study suffers from strong unrealistic assumptions on the distribution of noise and signal. The authors are aware of these limitations and conducted a simulation study to investigate their effect. However, the conducted simulated study in not well-designed with respect to actual realistic scenarios. I suggest:

- to simulate data with a more salient spatio-temporal structure (the covariance matrix of the simulated data is almost diagonal in Figure 2) in both signal and noise components (of course different structures for signal and noise).

**Main Review:**

*** Main Comments ***
- This paper is addressing an interesting and significant problem in the neuroimaging context.
- The idea of using the specifications of the Kronecker product in disentangling the spatio-temporal structure of the covariance matrix is interesting, however, this idea is not original and was presented in previous studies for the purpose other than MEEG source reconstruction.
- The text is clear but dense in mathematical notations and representations.
- This work's significance is limited by the strong (restrictive and wrong) assumptions that are made on 1) the independence of the sources across different brain locations, 2) identical temporal distribution for signal and noise. The authors are aware of these practical limitations and conducted a simulated study in a limited scenario arguing their negligible effect on the outcoming results.

*** Other Comments ***
- The authors categorize the problem in Eq. 1 as a multi-task regression problem, while originally the addressed problem diverges from the definition of multi-task learning in which the aim is to estimate $\mathbf{L}$ and not $\mathbf{X}$ (unlike brain source reconstruction).
- I would call $\Gamma$ as spatial and $\mathbf{B}$ as temporal covariance matrix because even though the $\Gamma$ is computed on time-series it at the end represents the spatial structure of the signal across different brain locations (and the same for $\mathbf{B}$).
- According to the ground-truth covariance matrix in Figure 2, the simulated signal can barely be considered structured. I suggest simulating data with a more salient structure in the signal and noise components to better evaluate the violation of assumptions.
- Despite considering the spatial structure of data, I wonder why the reconstructed sources are so sparse in Figure 3. I would expect a more smooth localized effect.
- It is not clear to me in what sense the proposed model is hierarchical. In the Bayesian context, hierarchical models have a specific definition (using the same prior in a hierarchy over a group of parameters).

**Time Spent Reviewing:**

3

---

> ### Author Response · Authors · 2021-08-10
> **Response to Reviewer bAFS**
>
> ## Responses to “Summary” and  “Main Comments”:
>
> We thank the reviewer for a concise summary of the paper and constructive and useful comments as well as helpful feedback. Below, we provide step-by-step responses to the reviewer’s comments:
>
> * We acknowledge studies in other domains have proposed the use of Kronecker product for spatio-temporal covariance structures. But to the best of our knowledge, no study has combined this effort along with the use of the majorization-minimization (MM) framework to derive efficient reconstruction algorithms.
>
> * Regarding assumptions made in the paper, they are all embodied as priors and since we are doing Bayesian inference, the posterior allows for estimation of data even if these priors are incorrect or inaccurate. Indeed our results both in simulations and real data show significant improvements in accuracy over benchmarks even when some of these assumptions are relaxed.
>
>    The assumption of independence of sources across brain locations is a ubiquitous assumption for many algorithms in the literature. In our case, we assume independence that is conditional on a spatio-temporal covariance, which results in Type-II reconstruction algorithms. This is to be contrasted with other Type-I methods which make more explicit factorial-independence assumptions across brain locations. Furthermore, the Dugh algorithm can be easily extended to other published models with hierarchical correlations among brain locations such as the multiple sparse priors (MSP) method [Friston et al., NeuroImage, 2008] or Tree Champagne algorithm [Cai et al., NeuroImage, 2018].
>
>    Regarding the assumption of identical temporal distributions for signal and noise, we refer to our response to a similar comment by Reviewer “exED”. In summary, we have shown (in Figure 1) that our algorithm performance is robust for empirical data that deviates from this assumption (AR model for sources and I.I.D model for noise), and we will be glad to conduct additional experiments with different spectra for source and noise to further demonstrate the robustness of our proposed method even if this assumption is violated. Besides, it is reasonable in many empirical scenarios to assume that the noise and source activity may share similar temporal covariances. For instance if the noise arises from other brain sources, they may share the same temporal covariances. However, if the noise arises from biological (eye-blink, EKG, EMG) or non-biological artifacts (electronic implants) they may not share the same temporal covariance structure and this assumption is relaxed. For such scenarios, following the ideas proposed by [Rakitsch et al., NeurIPS, 2013] and [Wu et al., NeurIPS, 2018], in future work, we believe that it is possible to obtain inference algorithms by exploiting eigenvalue decompositions of the sum of Kronecker products for spatial and temporal covariances. However, these inference procedures are significantly challenging both theoretically (in terms of proving convergence and obtaining closed-form update rules) and computationally,  with as yet unclear advantages in terms of the actual reconstructions obtained.
>
> ## Responses to “Other Comments”:
>
> * Our definition of multi-task regression problems is consistent with prior publications on this problem, for both MEG/EEG source reconstructions and fMRI ([Rakitsch et al., NeurIPS, 2013], [Massias et al., NeurIPS, 2018], [Xiao et al., TBME, 2019], [Bertrand et al., NeurIPS, 2019], and [Chevalier et al., NeurIPS, 2020]).
>
> * We agree with the reviewer's comment about the definitions of the spatial and temporal covariances, and the paper reflects this.
>
> * We would like to highlight that the model covariance we used was indeed for the most general cases (modeling a non-Toeplitz structure with random PSD covariance matrices), which is actually the more difficult problem and shows performance on this hard case. We agree with the reviewer that our choice of modeling the time series with completely random covariance matrices resulted in covariance models without more intuitive temporal neighborhood smoothness. In additional experiments, we would be happy to demonstrate performance for such more “intuitive” cases with smooth temporal covariance structure but that is also non-Toeplitz (e.g. having more visually distinct off-diagonal structure in the temporal covariance matrix).
>
> * We are not sure why the reviewer expects a more smooth localized effect. Our algorithm is designed to reconstruct spatially sparse sources with arbitrary temporal covariance structure, and this is consistent with our results. In figure 3, we show reconstruction of visual evoked fields under extremely low SNR conditions. These visual evoked responses are expected to reflect sparse activity arising from visual cortices for reconstructions at a 8 mm resolution spanning the whole brain even in the presence of large background brain activity. In this context, the spatial structure of the data can be considered to be sparse and the results are quite consistent with this assumption.
>
> * Our model is hierarchical because we have parameters and hyper-parameters estimated from data. Indeed as the reviewer points out the prior structure is the same, and the temporal covariance is assumed to be shared across sources.
>
> ## Response to “Limitations and Societal Impact”:
> We would like to highlight that the model covariance we used was indeed for the most general cases (modeling a non-Toeplitz structure with random PSD covariance matrices), which is actually the more difficult problem and show performance on this hard case. We agree with the reviewer that our choice of modeling the time series with completely random covariance matrices resulted in covariance models without more intuitive temporal neighborhood smoothness. In additional experiments, we would be happy to demonstrate performance for such more “intuitive” cases with smooth temporal covariance structure but that is also non-Toeplitz (e.g. having more visually distinct off-diagonal structure in the temporal covariance matrix).
>
>
> ### References:
>
> Friston, Karl, et al. "Multiple sparse priors for the M/EEG inverse problem." NeuroImage 39.3, 1104-1120, 2008.
>
> Cai, C., Sekihara, K., & Nagarajan, S. S., “Hierarchical multiscale Bayesian algorithm for robust MEG/EEG source reconstruction”. NeuroImage, 183, 698-715, 2018.
>
> Barbara Rakitsch, Christoph Lippert, Karsten Borgwardt, and Oliver Stegle. “It is all in the noise: Efficient multi-task Gaussian process inference with structured residuals”.  In Proceedings of the 26th International Conference on Neural Information Processing Systems-Volume 1, pages 1466–1474, 2013.
>
> Anqi Wu,  Stan Pashkovski,  Sandeep R Datta,  and Jonathan W Pillow.   “Learning a latent manifold of odor representations from neural responses in piriform cortex”. Advances in Neural Information Processing Systems, 31:5378–5388, 2018.
>
> Mathurin Massias, Olivier Fercoq, Alexandre Gramfort, and Joseph Salmon.   “Generalized concomitant multi-task lasso for sparse multimodal regression.” In International Conference on Artificial Intelligence and Statistics, pages 998–1007, 2018.
>
> Xiao, L., Stephen, J. M., Wilson, T. W., Calhoun, V. D., & Wang, Y. P., “A manifold regularized multi-task learning model for IQ prediction from two fMRI paradigms”. IEEE Transactions on Biomedical Engineering, 67(3), 796-806,  2019.
>
> Quentin Bertrand, Mathurin Massias, Alexandre Gramfort, and Joseph Salmon.   “Handling correlated and repeated measurements with the smoothed multivariate square-root Lasso.”  In Advances in Neural Information Processing Systems, pages 3959–3970, 2019.
>
> Jérôme-Alexis Chevalier, Alexandre Gramfort, Joseph Salmon, and Bertrand Thirion. “Statistical control for spatio-temporal MEG/EEG source imaging with desparsified multi-task lasso.” In Advances in Neural Information Processing Systems, 2020.

---

### Official Review · Reviewer_txKn · 2021-07-15

**Rating:** 7
**Confidence:** 3

**Summary:**

This paper proposes an efficient method for source reconstruction from M/EEG data that estimates the spatiotemporal correlation structure in both the source space and the measurement error. The spatiotemporal pattern of the source and noise covariances is given by a Kronecker product structure with an independent (diagonal) spatial structure and either a full or Toeplitz temporal structure, where the temporal component is shared across the source and noise components. Since the full optimization problem is non-convex, the authors propose a majorization-minimization (MM) algorithm with provable convergence and derive simple, efficient iterative updates for each component. Two versions of the model are proposed: full Dugh, which uses a full positive definite matrix for the temporal covariance; and thin Dugh, which can be applied in the stationary data setting and uses a Toeplitz temporal covariance structure with an alternative set of efficient updates for the MM algorithm. On simulated data with known sources, thin Dugh achieves comparable or better performance as compared to existing source reconstruction methods (Champagne, eLORETA, and S-FLEX). When applied to simulated data that was generated while varying the true underlying temporal covariance structure, full Dugh is shown to outperform thin Dugh when the true matrix is full and vice versa when it has Toeplitz structure. Finally, experiments on real MEG data from subjects presented with visual or auditory stimuli show that full and thin Dugh can recover plausible and well-localized temporal and spatial patterns for the reconstructed sources, even with a very limited number of trials.

**Ethical Concerns:**

No ethical issues come to mind.

**Limitations And Societal Impact:**

The authors have discussed the technical limitations of their work, but not the potential negative societal impact. The most salient potential impact of this work is its use in neuroscience research, where any potential technical biases or shortcomings of the method could be reflected in the findings of any researcher who applies this method to their data. The authors could briefly discuss this potential impact in the final section of the paper. Beyond this, there are no obvious harmful societal consequences of this work to my knowledge.

**Main Review:**

Overall, the paper is clearly written. The presentation of the method is sound, there is enough technical detail to understand the model and algorithm, and the experiments do a sufficient job of demonstrating the Dugh method's utility both when compared to other source estimation approaches on simulated data and for recovering plausible localized activity on real MEG data. I particularly appreciated the comparison between thin Dugh and full Dugh, both on the real MEG data as well as on synthetic data with known structure to demonstrate that each version of the method excels when the underlying structure aligns with exactly what it was designed for.

One issue I had with the paper was the limited discussion of and comparison to other source imaging methods that account for temporal structure in some way. This could include approaches such as mixed-norm inverse methods, which do encourage some temporal smoothness in the estimated source activations even if they don't model a full temporal covariance structure as the proposed method does. While such other methods are briefly referred to in the paper, comparing to other approaches that model temporal structure could further showcase the particular strength of the Kronecker-factored covariance structure and associated optimization algorithm that the authors present.

Another interesting experiment would have been a runtime comparison between the proposed method and the other methods used in the experiments. The authors claim their algorithm is efficient and derive a set of simple closed-form updates, but no direct comparison of efficiency is made against the other source estimation methods. In particular, this would validate the point the authors make about other methods that model temporal structure having "computationally over-demanding inference schemes" (line 6). A runtime vs. accuracy comparison to other spatio-temporal methods as well as the baselines already included in the paper would strengthen the presentation of the proposed Dugh method as an accurate source estimation method that is also less computationally intensive compared to alternatives that also model temporal structure.

Other minor comments:
- The caption for Figure 2 mentions "Toeplitz structure in the first row, and full covariance structure in the second row", when the figure shows the reverse

**Time Spent Reviewing:**

3

---

> ### Author Response · Authors · 2021-08-10
> **Response to Reviewer txKn**
>
> ## Response to the “Summary” and “Main Review”:
> We thank the reviewer for a concise summary of the paper and constructive and useful comments as well as helpful feedback.
>
> Regarding the raised issue by the reviewer about benchmark comparison with other source imaging methods that account for temporal structure, we would like to state that, indeed, it is possible to include temporal derivative(s) constraints within a mixed-norm framework to impose temporal smoothness without explicit modeling of temporal covariance structure. Using first-temporal-derivative L2-norm constraints would impose smoothness constraints while second-temporal-derivative L1-norm constraints would impose piecewise linear sparse temporal shaping of reconstructions. And Wavelet or Gabor transform-based constraints could be employed for arbitrary time-frequency shaping ([Gramfort et al., NeuroImage, 2013], [Strohmeier et al., PRNI, 2015], and [Strohmeier et al., IEEE TMI, 2016]). These methods, in general, will require the estimation of multiple regularization parameters by cross-validation. Besides, published algorithms in this area have mainly used these constraints to enforce sparsity, but they can be used to indirectly learn temporal covariance structure, and as such, there are no standard methods in the field for this approach.
>
> In contrast, our approach automatically models general temporal covariance structure and allows for reconstructing arbitrary temporal structure (high-pass, band-pass, etc) without the use of arbitrary constraints. If provided an opportunity to revise, we would be happy to include such comparisons with benchmarks.
>
> Regarding runtime comparisons of accuracy and efficiency of the convergence of loss-function, we would certainly be glad to include some information either in the supplementary materials. We have also modified the text in the abstract to revise the following sentence: "computationally over-demanding inference schemes" accordingly.
>
> ## Response to “Other minor comments”:
> Thank you for spotting that typo. We will fix it in revision.
>
> ## Response to “Limitations and Societal Impact”:
> We are solving inverse problems that have non-unique solutions which depend on explicit or implicit assumptions of priors. These have applications in basic, clinical, and translational neuroscience imaging studies. Our use of a hierarchical empirical Bayesian framework allows for explicit specifications of the priors in our model that are learned from data, and a more clear interpretation of our reconstructions with reduced bias. Nevertheless, in our algorithms, we assume sparse spatial priors, and in scenarios where this assumption embodied in the priors is incorrect, the resulting reconstructions will be inaccurate. Users will need to be cognizant of these issues. We would be happy to add statements that reflect these points in a revision.
>
>
> ### References:
>
> Gramfort, A., Strohmeier, D., Haueisen, J., Hämäläinen, M. S., & Kowalski, M.,  “Time-frequency mixed-norm estimates: Sparse M/EEG imaging with non-stationary source activations”. NeuroImage, 70, 410-422, 2013.
>
> Strohmeier, D., Gramfort, A., & Haueisen, J., “MEG/EEG source imaging with a non-convex penalty in the time-frequency domain”. In 2015 International Workshop on Pattern Recognition in NeuroImaging (pp. 21-24). IEEE. 2015.
>
> Strohmeier, D., Bekhti, Y., Haueisen, J., & Gramfort, A. “The iterative reweighted mixed-norm estimate for spatio-temporal MEG/EEG source reconstruction”. IEEE transactions on medical imaging, 35(10), 2218-2228, 2016.

---

> > ### Comment · Reviewer_txKn · 2021-08-29
> > **Comment on author response**
> >
> > Thank you for your detailed response. I appreciate the willingness to include a comparison to other methods that account for temporal correlation structure, and I definitely think this would strengthen the empirical results. However, the current set of experiments seems sufficient to me for showcasing the capabilities of the proposed approach and comparing to other methods used in practice. The discussion of "Limitations and Societal Impact" outlined above is reasonable, and should be added to the main text. Overall, the authors' response adequately addresses my concerns, and I am happy to maintain my rating.

---

### Official Review · Reviewer_exED · 2021-07-18

**Rating:** 7
**Confidence:** 4

**Summary:**

Inverse modeling is at the heart of medical imaging techniques, especially in neuroimaging. Many techniques have been developed in recent years, first based on PDE techniques and, more recently, by machine learning approaches. The addition of a priori knowledge to these techniques allows a better understanding of the studied phenomenon. Therefore, it is necessary to develop robust methods combining modeling, a priori knowledge, while controlling the computational cost. This article proposes to derive an efficient inference algorithm with spatio-temporal dynamics in model parameters and noise. The spatial and temporal duality is realized by using the product of
Kronecker product. They assume sparse spatial covariance matrices, while the temporal covariance is modeled to have either full or Toeplitz structure. They then define a Bayesian model whose parametric estimation is performed by a Maximization-Minimization algorithm. The proposed results are motivated by demonstrations (detailed in appendix) and numerical experiments.

**Ethical Concerns:**

I did not identify any ethical issues in this document. A call for volunteers was required for this work. Participants had to provide written informed consent and received monetary compensation.

**Limitations And Societal Impact:**

The article discusses methods to improve medical imaging techniques. Unless medicine is misused, it seems to me that it cannot have a negative social impact (but I could be wrong).

**Main Review:**

The article is well constructed, as are the appendices, which provide many insights.

The demonstrations, postponed to the appendix, are pleasant to follow. Moreover, for those I have reread, at least, they seem correct unless I am mistaken. However, they are essentially based on Riemannian geometry, which may reduce their scope. The authors have nevertheless made an effort to give the essential definitions in appendix C (even if I doubt that this is sufficient for a "neophyte" in this field).
I particularly appreciated that each theorem stated in the article is precisely referred to its proof.

The chosen approach seems innovative. However, I wonder what sense it makes to assume the same temporal correlation for noise and sources in terms of modeling. And to what extent is this assumption realistic, encountered in practice, reasonable? I am aware that this makes the computation easier, as claimed by the authors in the discussion. Are the authors convinced that this assumption can also be relaxed theoretically and not just empirically as is currently the case?

Small remarks: prefer to use a git repository rather than a folder on a dropbox, which seems to me less perennial.
Also, Figure 1 is not readable when printed in black and white: I suggest that authors use different symbols for each node of the curves and include them in the legend.

**Time Spent Reviewing:**

5

---

> ### Author Response · Authors · 2021-08-10
> **Response to Reviewer exED**
>
> ## Response to the “Summary” and “Main Review”:
>
> We thank the reviewer for a concise summary of the paper and constructive and useful comments as well as helpful feedback.
>
> Regarding the assumption of the same temporal correlation for the sources and noise, as noted by the reviewer, this does help reduce the computational complexity of the problem. Nevertheless, we have shown (in Figure 1) that our algorithm performance is robust for empirical data that deviates from this assumption (AR model for sources and I.I.D model for noise). We will be glad to conduct additional experiments with different spectra for source and noise to further demonstrate the robustness of our proposed method even if this assumption is violated.
>
> In terms of empirical evidence for this assumption, it is reasonable in many scenarios to assume that the noise and source activity may share similar temporal covariances. For instance, if the noise arises from other brain sources, they may share the same temporal covariances. However, if the noise arises from biological (eye-blink, EKG, EMG) or non-biological artifacts (electronic implants), the source and noise may not share the same temporal covariance structure, and this assumption is relaxed. Even though we did not intend for Dugh to serve as an artifact rejection method, our empirical findings of the robustness of reconstructions for different spectra for source and noise suggest that it could serve that purpose as well, and we will be happy to conduct some additional experiments to support this claim.
>
> For generative models where the noise and source spectra are not shared, by following the ideas proposed by [Rakitsch et al., NeurIPS, 2013] and [Wu et al., NeurIPS, 2018], in future work, we believe that it is possible to obtain inference algorithms by exploiting eigenvalue decompositions of the sum of Kronecker products for spatial and temporal covariances. However, these inference procedures are significantly challenging both theoretically (in terms of proving convergence and obtaining closed-form update rules) and computationally, with as yet unclear advantages in terms of the actual reconstructions obtained.
>
> ## Response to “small remarks”:
> Regarding using Dropbox rather than GitHub, we did not share our GitHub repository to ensure anonymity during the review process, but we will be glad to include that in a final version. We will also be glad to modify figure 1 by incorporating the reviewer's suggestion to ensure readability in black and white.
>
> ### References:
> Barbara Rakitsch, Christoph Lippert, Karsten Borgwardt, and Oliver Stegle. “It is all in the noise: Efficient multi-task Gaussian process inference with structured residuals”.  In Proceedings of the 26th International Conference on Neural Information Processing Systems-Volume 1, pages 1466–1474, 2013.
>
> Anqi Wu,  Stan Pashkovski,  Sandeep R Datta,  and Jonathan W Pillow. “Learning a latent manifold of odor representations from neural responses in piriform cortex”. Advances in Neural Information Processing Systems, 31:5378–5388, 2018.

---

### Decision · Program_Chairs · 2021-09-27

**Decision:**

Accept (Poster)

**Comment:**

The authors present an efficient method to infer models with both spatial and temporal correlations by leveraging Kronoker, sparse, and toeplitz structures. The authors found the method interesting and highly practical as a way to bring additional structure into the regression framework without undue computational burden. While the reviewers did raise a few minor points related to the restrictive model assumptions and the potential for additional comparisons, the authors have provided thorough responses that the reviewers have found adequate. Thus I am happy to recommend this work for acceptance to NeurIPS.